JCB Journal of Cell Biology

TOOLS

# High-throughput ultrastructure screening using electron microscopy and fluorescent barcoding

Yury S. Bykov[1,2]*, Nir Cohen[3]*, Natalia Gabrielli[1], Hetty Manenschijn[1,4], Sonja Welsch[1], Petr Chlanda[1], Wanda Kukulski[1,4], Kiran R. Patil[1], Maya Schuldiner[3], and John A.G. Briggs[1,2,4]

Genetic screens using high-throughput fluorescent microscopes have generated large datasets, contributing many cell biological insights. Such approaches cannot tackle questions requiring knowledge of ultrastructure below the resolution limit of fluorescent microscopy. Electron microscopy (EM) reveals detailed cellular ultrastructure but requires time-consuming sample preparation, limiting throughput. Here we describe a robust method for screening by high-throughput EM. Our approach uses combinations of fluorophores as barcodes to uniquely mark each cell type in mixed populations and correlative light and EM (CLEM) to read the barcode of each cell before it is imaged by EM. Coupled with an easy-to-use software workflow for correlation, segmentation, and computer image analysis, our method, called "MultiCLEM," allows us to extract and analyze multiple cell populations from each EM sample preparation. We demonstrate several uses for MultiCLEM with 15 different yeast variants. The methodology is not restricted to yeast, can be scaled to higher throughput, and can be used in multiple ways to enable EM to become a powerful screening technique.

## Introduction

Functional studies can be extended from individual proteins to a proteome-wide level using high content screens relying on genetic tools, fluorescent light microscopy (LM), and automated workflows. Budding yeast (*Saccharomyces cerevisiae*, hereafter referred to simply as yeast) is a widely used model organism for such high-throughput studies. Easy and scalable genetic manipulation has allowed the creation of tools such as systematic deletion libraries and GFP collections (Giaever et al., 2002; Huh et al., 2003; Yofe et al., 2016; Weill et al., 2018). Combined with automatic fluorescence microscopy, these systematic libraries help to address a large variety of questions in cell biology (Ohya et al., 2005; Cohen and Schuldiner, 2011; Breker et al., 2014). Some questions cannot be addressed or solved at the resolution limit of LM, but require higher resolution techniques such as EM. However, EM has suffered, until now, from very low throughput.

With the introduction of fully computer-controlled electron microscopes, it is now possible to perform automated and large-scale data collection. This has particularly benefited the field of 3D EM, which relies on the collection of a large number of projections (tomography) or serial sections (Zheng et al., 2004; Mastronarde, 2005; Suloway et al., 2005; Peddie and Collinson, 2014; Schorb et al., 2019). The number of individual samples that can be analyzed by EM, however, remains relatively low because sample preparation procedures include time-consuming manual steps, and because samples are typically inserted individually into the electron microscope. To obtain the best preservation of both ultrastructure and fluorescence in yeast, each sample is subjected to high-pressure cryofixation, freeze substitution, manual sectioning using a microtome, mounting on EM grids, insertion into the electron microscope, and, of course, visualization and image analysis (McDonald, 2007; McDonald et al., 2010). These manual procedures have prevented application of the high-throughput screening paradigms that are common in LM to the ultrastructural features that can be observed only by EM.

Here, we present an in-resin correlative light and EM (CLEM) approach (Kukulski et al., 2011; Spiegelhalter et al., 2014; Bykov et al., 2016) to increase the throughput of EM experiments (Fig. 1 A). We apply it to yeast, but the approach can be adapted to other

[1]Structural and Computational Biology Unit, European Molecular Biology Laboratory, Heidelberg, Germany; [2]Structural Studies Division, Medical Research Council Laboratory of Molecular Biology, Cambridge Biomedical Campus, Cambridge, UK; [3]Department of Molecular Genetics, Weizmann Institute of Science, Rehovot, Israel; [4]Cell Biology and Biophysics Unit, European Molecular Biology Laboratory, Heidelberg, Germany.

*Y.S. Bykov and N. Cohen contributed equally to this paper; Correspondence to John A.G. Briggs: john.briggs@mrc-lmb.cam.ac.uk; Maya Schuldiner: maya.schuldiner@weizmann.ac.il; Y.S. Bykov's present address is Department of Molecular Genetics, Weizmann Institute of Science, Rehovot, Israel; H. Manenschijn's present address is Department of Biochemistry and NCCR Chemical Biology, University of Geneva, Geneva, Switzerland; S. Welsch's present address is Thermo Fisher Scientific, Eindhoven, Netherlands; P. Chlanda's present address is BioQuant, University of Heidelberg, Heidelberg, Germany; W. Kukulski's present address is Cell Biology Division, Medical Research Council Laboratory of Molecular Biology, Cambridge Biomedical Campus, Cambridge, UK.

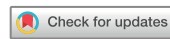

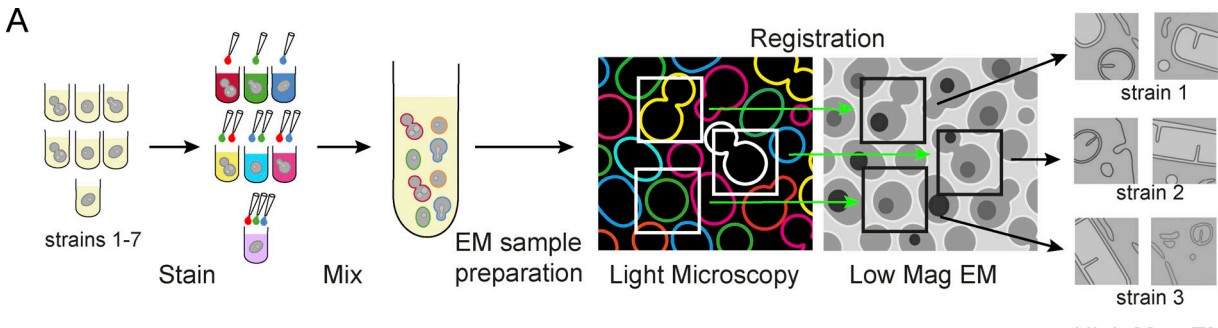

**Figure 1. The principle of fluorescent barcoding CLEM. (A)** Schematic of MultiCLEM workflow: strains or cells in different experimental conditions are grown in parallel, and each population is stained with a unique combination of Con A conjugates (figure demonstrates three colors, but more can be used) to produce a combination (the barcode); strains are then mixed together and processed for EM. Samples are imaged with LM and EM at medium magnification (Mag), correlation is performed, cell identities are determined using the fluorescent barcode, and the high-resolution information is collected for each strain. **(B)** An example using three fluorophores to give seven well-discriminated combinations: pseudocolor composite image of a 100-nm-thick Lowicryl section of embedded cells labeled with Con A conjugated to Alexa 350, Alexa 488, and TRITC. **(C)** Examples of individual cells with all possible barcodes; each column corresponds to one fluorescent channel: R, TRITC; G, Alexa 488; B, Alexa 350. Scale bars: 10 μm in B, 2 μm in C.

cell types. First, cells from different strains or genetic backgrounds or under different experimental conditions are grown in parallel. Then, each of them is treated by a unique combination of fluorescent labels, creating a barcode. The barcode provides a specific staining pattern that marks the identity of each cell but does not influence its genotype or physiology. Following labeling, cells from parallel experiments are mixed together and undergo a single, unified, EM sample preparation. Prior to EM, the sample is imaged by LM to obtain the barcodes (staining patterns) for each cell. Correlation is performed between fluorescent and EM images, and cell positions and identities are determined. By multiplexing both sample preparation and EM imaging, this method substantially increases potential throughput. Furthermore, it removes the variability inherent in EM preparation of separate samples, allowing direct, automated comparison of images from parallel experiments and accurate quantification of traits.

We call our new methodology MultiCLEM (for multiplexed CLEM). This methodology opens the door to new possibilities in cell biology where ultrastructural questions can be asked at a throughput previously only available for gross morphological changes in the cell.

## Results

### Fluorescent labeling of the cell wall enables molecular barcoding

Sample barcoding is a common parallelization approach in biology (Krutzik and Nolan, 2006; Smith et al., 2009; Knapp et al., 2012), and combinatorial fluorescent labeling is a powerful way to discriminate objects within heterogeneous samples (Livet et al., 2007; Valm et al., 2012). We therefore decided to harness these approaches to multiplex biological sample preparation for EM. We selected fluorophores that are biologically inert, are very bright, retain their fluorescent signal after EM sample preparation, stain the same compartment consistently in different fluorescent channels, and can be easily delivered to the stained compartments in a combinatorial way. Fluorescent conjugates of Con A that stain the yeast cell wall fulfilled these requirements. Con A conjugates in multiple colors are commercially available, and additional dyes can be easily attached using various protocols (Toseland, 2013). We selected five Con A conjugates (colors) that can be resolved on many conventional wide-field microscopes: Alexa 350 (blue), Alexa 488 (green), TRITC (orange), Alexa 647 (far red), and Cy7 (near infrared).

To obtain specific barcodes, the selected Con A conjugates are mixed with each other in all possible combinations. Each particular conjugate (color) can be either present or absent in the mix (barcode). The total number of conjugates used depends on the number of available fluorescent channels in the LM. More conjugates allow for more barcodes (Fig. 1 A). To increase the accuracy of barcode determination, we did not use the possibility to mix Con A conjugates in different ratios and also excluded the combination with no colors present to avoid false negatives. This gives maximum 7 ($2^3 - 1$), 15 ($2^4 - 1$), and 31 ($2^5 - 1$) barcodes for three, four, and five Con A dyes, respectively.

We optimized dye mix preparation and staining conditions to achieve the bright and uniform cell wall staining necessary for automated image processing. This resulted in an easy-to-use barcoding protocol (see Materials and methods and supplemental data for complete protocol) that could be followed by the already established sample fixation, resin embedding, and LM imaging using a protocol that optimizes both preservation of fluorescent signals and cellular ultrastructure (Kukulski et al., 2012).

To illustrate the barcoding principle, we prepared a set of yeast cultures using three Con A conjugates (Alexa 350, Alexa 488, and TRITC) combined in seven possible ways as described above. These cultures were mixed together in a single sample, which was subjected to EM sample preparation and LM (Fig. 1, B and C). When the three fluorescent channels are displayed as a pseudocolor overlay of red, green, and blue, it is easy for the human eye to distinguish the seven possible combinations of three primary colors (red, green, blue, cyan, magenta, yellow, white), and thereby identify the source culture for each cell (Fig. 1 C).

We next verified that the barcodes were not corrupted during sample preparation through leaking of dyes to adjacent cells. To do this, we stained a strain expressing cytosolic GFP with Con A conjugated to Alexa 647 and a strain expressing mCherry with Con A conjugated to Alexa 350 (Fig. S1). We mixed these cultures together and incubated them in a pellet for 15, 30, or 60 min to visualize whether there is a time-dependent exchange of Con A between different cells. Then we imaged the sample immediately or after EM sample preparation. We found that, regardless of time or visualization method, <0.3% of cells had acquired the second cell wall stain (Fig. S1; see Materials and methods for a detailed description of the analysis). Moreover, those cells that did acquire the second dye due to proximity to other cells had increased autofluorescence, suggesting that they were dead or sick, which would have excluded them from further analysis at later steps anyway. Since our sample preparation protocol typically requires <5 min of the cells residing in a pellet, we conclude that corruption of the barcode during sample preparation is very unlikely.

Manual analysis of images for sample identification is time-consuming and restricts expansion of this methodology to high-throughput applications. We therefore developed a workflow for automation of barcode reading, correlation, and targeting of automated high-resolution acquisition using Matlab and SerialEM (Mastronarde, 2005; Schorb et al., 2019; Fig. 2). The workflow is organized in a Matlab graphic user interface. First,

the user assists correlation of medium magnification EM data with LM data, then barcodes and coordinates of all cells are automatically determined, and finally coordinates of cells are imported back to the electron microscope. SerialEM then uses these coordinates for automated high-resolution EM imaging (for detailed workflow description, see Materials and methods; the code and associated documentation is available at https://github.com/ybyk/muclem, https://www2.mrc-lmb.cam.ac.uk/groups/briggs/resources, and http://mayaschuldiner.wixsite.com/schuldinerlab/multiclem; Briggs Group, 2019; Schuldiner Group, 2019). Together, our barcoding approach and the software associated with it allow automated image acquisition of hundreds of cells from multiple samples for image analysis and phenotypic profiling.

## MultiCLEM allows quantitative ultrastructural phenotyping

The advantage of our proposed approach is that it allows the study of ultrastructural phenotypes and the quantification of parameters inaccessible to LM, while reducing time spent on sample preparation and data acquisition. Pooling many strains together also reduces variation between samples due to the preparation procedure, enabling more reliable quantification of complex phenotypes. To assess the developed MultiCLEM pipeline, we chose to compare how the fine ultrastructure of membranous organelles is altered in different genetic backgrounds subjected to stress conditions.

We selected seven yeast strains: a reference laboratory strain S90 (Steinmetz et al., 2002); two different wine strains, PRICVV50 (Novo et al., 2009) and SFB2 (Padilla et al., 2016); and four deletion mutants from the prototrophic deletion library (Mülleder et al., 2012) with defects in stress response and endomembrane system organization, Δhog1, Δtcb1, Δvps35, and Δlpl1 (Hohmann et al., 2007; Liu et al., 2012; Manford et al., 2012; Selvaraju et al., 2014). All strains were grown in parallel and then either subjected to hyperosmotic shock (1 M sorbitol for 45 min) or not. The 14 resulting yeast cultures underwent MultiCLEM (Fig. 3 A) using combinations of four fluorophores.

All 14 barcodes were successfully recovered by the computational pipeline (Fig. S2). We first assessed the reliability with which cells were assigned to the correct strain. For 1,000 cells, we performed a careful visual assessment of the fluorescence signals to verify the barcode, generating a "gold standard" dataset that we compared with automated assignment: 87% of cells were assigned to the same class visually and automatically, while 4% of cells were incorrectly assigned by the automated procedure. 8% of cells could not be classified visually due to the absence of cell wall staining or the signal obscured by aggregated Con A, and thus automatic assignment was random. While such a low false-positive and -negative rate may be acceptable for high-throughput analysis, there are some situations where it is not acceptable. For such cases where higher accuracy is desired, we created an easy tool for quality controlling and perfecting assignments. Our tool enables examination of the data for each cell so that both the barcode assignment and EM preservation quality can be visually assessed at the same time. The user can then manually correct the barcode assignment or exclude cells from the dataset based on ambiguous barcode or poor EM image quality (see Materials and methods for

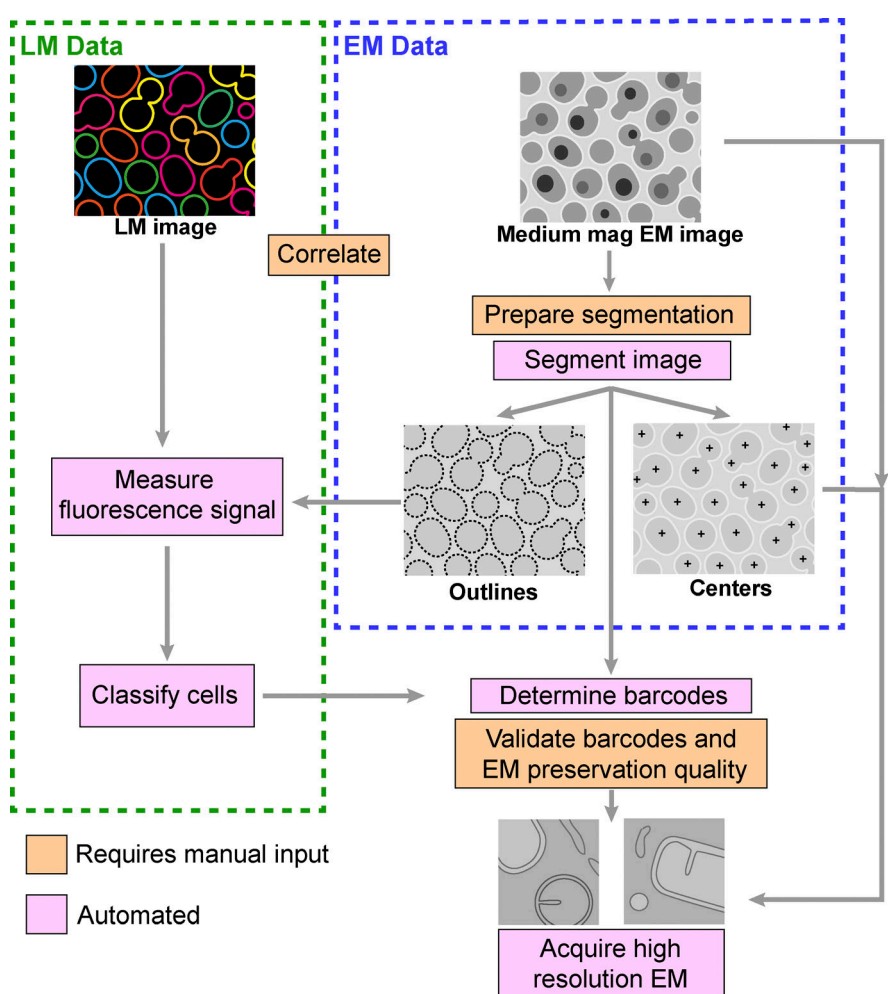

**LM Data**

**EM Data**

LM image

Medium mag EM image

Correlate

Prepare segmentation

Segment image

Measure fluorescence signal

Outlines

Centers

Classify cells

Determine barcodes

Validate barcodes and EM preservation quality

Requires manual input

Automated

Acquire high resolution EM

Figure 2. **MultiCLEM data processing pipeline.** A schematic overview of the data processing pipeline. Medium-magnification EM images are segmented to derive cell centroids and outlines. Correlation of EM and LM data is performed, after which cell outlines determined from the EM data are used to create masks to measure fluorescence intensity. Cells are classified by fluorescence intensities, and barcodes are determined. Cells with unknown barcode or poor preservation can be manually excluded during a quality control step (see Materials and methods). Centroids of selected cells are imported back to the electron microscope for high-resolution imaging. The processing steps requiring some manual input are marked in orange, and fully automated steps are marked in pink.

details). This tool enables assessment of hundreds of cells in one hour. After visual assessment, the user can perform high-resolution EM only on the cells that passed the quality control. The code and associated documentation is available at https://github.com/ybyk/muclem, https://www2.mrc-lmb.cam.ac.uk/groups/briggs/resources, and http://mayaschuldiner.wixsite.com/schuldinerlab/multiclem (Briggs Group, 2019; Schuldiner Group, 2019).

We then proceeded with high-magnification data collection. Previously assigned cells could be identified for high-magnification EM imaging in a precise and error-free way using SerialEM version 3.7 (Mastronarde, 2005; Schorb et al., 2019). Once set up, high-magnification imaging can run for a few days fully automatically. On different instruments, high-magnification data collection speeds ranged from 30 to 70 cell cross sections per hour at magnifications with pixel size ~1 nm. The maximal number of cells per strain that can be imaged is obviously dependent on data collection speed, length of data collection, and number of strains within the multiplexed sample, but >100 cell cross sections per strain can be attained routinely on standard EM setups. For the experiment described above, we acquired images of 1,748 cell sections from which we generated galleries of ~100 high-resolution micrographs per cell strain and experimental condition (Fig. S3).

The overall time needed to complete a MultiCLEM experiment is 2–3 wk, similar to that required for a single EM or CLEM experiment (Kukulski et al., 2012). This similarity is due to the fact that the timeline for a single EM or a MultiCLEM experiment is dominated by the freeze-substitution and resin-hardening step, which takes 1–2 wk in both cases. A small amount of work more than the usual EM process is required for fluorescence imaging (30 min per one grid), separate medium-magnification EM imaging (2–3 h per grid requiring only 15 min of user input), and computer work (assigning strains, performing correlations, and quality controlling) taking ~1 h per one grid square containing 200–300 cell cross sections. The increase in throughput of the most laborious steps was dramatic: 14 samples were sectioned simultaneously (1 h for one block instead of 14 h for 14 blocks). They were inserted into the microscope in one step and imaged in an automated manner in only two stages: a short session to acquire medium-magnification maps (2–3 h) and a long session for high-magnification imaging (~24 h). Since it is feasible to perform two to five such EM experiments in parallel during a 2–3-wk period, this makes it feasible to now study tens or even hundreds of strains where before only a few strains could be analyzed in a similar time period.

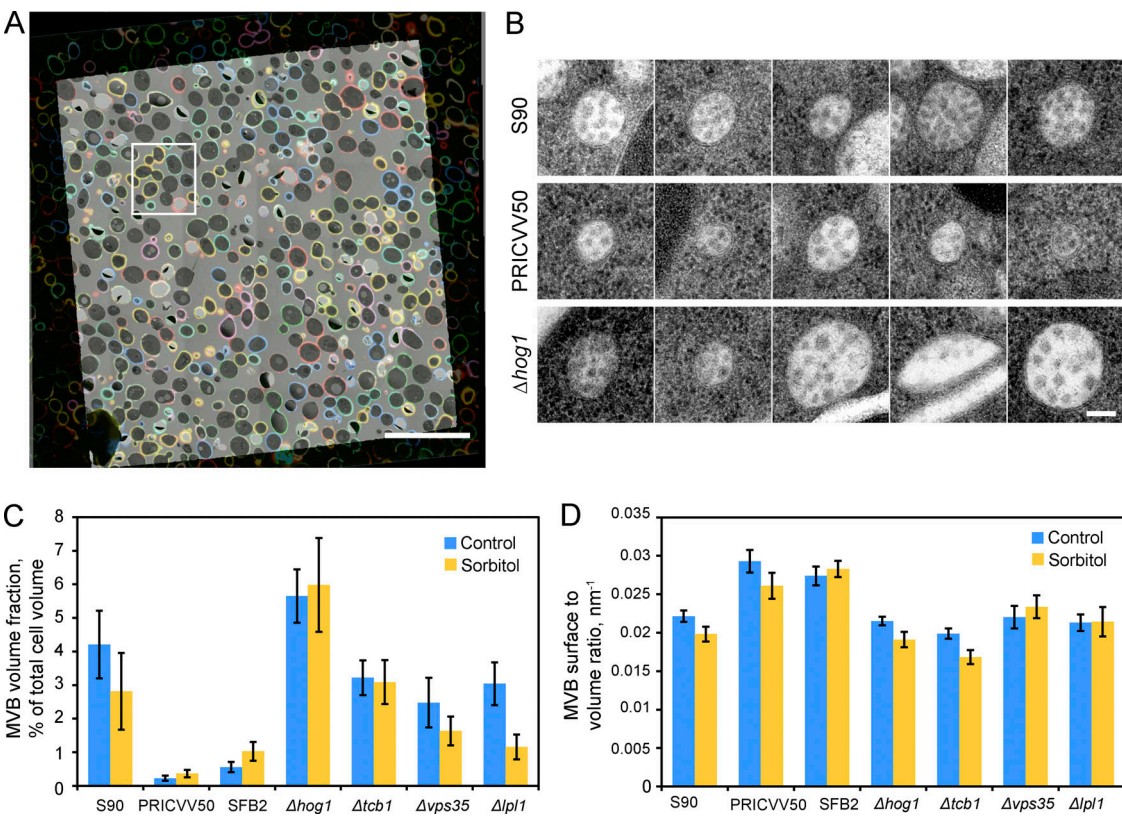

Figure 3. **Ultrastructural phenotyping of yeast with different genetic backgrounds subjected to osmotic shock. (A)** Overlay of pseudocolor composite fluorescent barcode image and medium-magnification EM map of one grid square of the sample represents how alignment is performed between the LM and EM. The region in the white rectangle is shown in detail in Fig. S2, A–D. **(B)** Representative MVB cross sections from the laboratory strain S90, wine isolate PRICVV50, and BY4741 with a Δhog1 mutation. **(C)** MVB volume fractions in total cell volume in different strains and conditions; error bars show SEM calculated as described in Materials and methods. **(D)** MVB outer membrane surface-to-volume ratio in different strains and conditions; error bars show SEM. In C and D, 510 MVBs in 1,748 cell cross sections were analyzed. Scale bars: 20 µm in A, 100 nm in B.

After confirming the effectiveness of our protocol, we examined the resulting high-resolution dataset. Yeast ultrastructure and preservation were similar to previously published data for healthy yeast cells in exponential growth phase that underwent high-pressure freezing and freeze substitution (Giddings et al., 2001; McDonald, 2007), meaning that yeast ultrastructure and preservation were not seriously affected by our barcoding protocol. Upon the switch to hyperosmotic conditions, yeast vacuoles become fragmented within a few minutes (LaGrassa and Ungermann, 2005). We did not observe any effect of changing the osmolarity of the medium on vacuole morphology, although it is possible that the vacuoles started fusing back during the barcoding step. While all EM embedding protocols may modulate ultrastructure to some extent, our approach optimizes the ability to compare strains, because the control strain is processed within the same multiplexed sample.

By visual inspection, for most organelles we observed no obvious morphology changes. However, striking differences were observed in the morphologies of mitochondria and multivesicular bodies (MVBs) compared to the reference strain (Figs. 3 B and S4 A). Mitochondria in the wine yeast strain SFB2 were swollen and lacked electron density in the matrix in both control and osmotic stress conditions (Fig. S4 A). We focused our attention, however, on the structural changes observed in MVBs.

We decided to use the possibility presented by EM that allows quantitative measurements of cellular features at the resolutions inaccessible to LM. Inspired by MVB morphology differences observed during visual examination of the dataset, we localized and measured 510 MVB cross sections in 1,748 imaged cell cross sections and estimated average volume fraction and surface-to-volume ratio of these organelles (Fig. S4, B and C; see Materials and methods for details). We found dramatic variation in the fraction of cellular volume taken up by MVBs (MVB volume fraction; Fig. 3 C), while the surface-to-volume ratios of MVBs were relatively uniform (Fig. 3 D). The reduced MVB volume fraction in SFB2 and PRICVV50 strains reflects both reduced abundance and reduced size of the MVBs (Fig. S4 D).

Yeast was one of the main model organisms used to uncover the mechanisms of MVB biogenesis, which involves fusion of individual endocytic vesicles and formation of intraluminal vesicles tightly controlled by Rab5 GTPases Vps21, Ypt52, and Ypt53 (Katzmann et al., 2001; Nickerson et al., 2010; Hanson and Cashikar, 2012; Russell et al., 2012; Arlt et al., 2015; Shideler et al., 2015). It was demonstrated that Ypt53 expression is induced under Ca²⁺ stress, and this might lead to increased numbers of MVBs and promote stress tolerance (Nickerson et al., 2012). However, variation of MVB size and abundance was not studied in detail in other growth conditions and yeast

life stages. Usually, with increase of cell size, including the transition from haploid to diploid, the fraction of cellular volume taken up by different organelles tends to stay constant or show a slight increase (Chan and Marshall, 2010). We observed that the diploid wine strains are larger and have dramatically reduced MVB volume fraction compared with the other strains (Figs. 3 C and S4 B). This is likely to reflect the different genetic background of the wine yeast strains: genome sequencing and phenotyping reported that strains of such origin differ significantly in their physiology from beer and laboratory yeast strains that have been cultured in rich media for many generations and, importantly, have lost many stress-response capabilities (Gallone et al., 2016). Our method can be useful for subsequent study of this topic and for exploration of other organelles similar to MVBs, whose volumes and sizes cannot be precisely measured using LM.

## MultiCLEM as part of a two-step screening strategy

The ultimate goal of MultiCLEM in yeast is to be used as a screening platform for cell traits not screenable with the standard techniques based on LM. At present, our setup cannot support the screening of an entire yeast deletion library, although future developments (see Discussion) could in principle bring it to a scale compatible with whole-genome screens. Hence a current, valuable utilization of MultiCLEM could be in secondary screens where the primary LM screen narrows down the number of strains and then a secondary screen to verify hits can be performed by MultiCLEM. As proof of principle of this prospective use, we performed a primary LM screen followed by a secondary MultiCLEM screen on a cellular phenotype not easily discernable by LM: mitochondrial ultrastructure.

The mechanisms underlying mitochondrial fission and fusion have been well worked out in yeast and are highly regulated and interrelated processes (Friedman and Nunnari, 2014). Surprisingly, however, it is still not clear how mitochondrial width and branching are regulated or determined. Moreover, even in the complete absence of both fission and fusion machineries, mitochondria still display differing morphologies, branching, and dynamics, suggesting that additional mitochondrial shaping proteins exist. To uncover such proteins, we performed a two-stage screen consisting of a higher-throughput live-cell LM stage to select initial hits followed by a lower-throughput high-resolution characterization of the ultrastructure of these hits using MultiCLEM.

Since we hypothesize that a protein important for mitochondrial shaping should reside in the mitochondrial outer membrane, in the first stage we imaged mitochondria labeled with endogenous Tom20 fused with GFP in 99 strains each overexpressing one mitochondrial protein, mostly outer membrane proteins. We performed this step on the background of a deletion in the yeast dynamin-like protein Dnm1, in whose absence the major fission events cannot take place (Fig. 4 A). Δdnm1 mutant cells show dense, clumped mitochondrial networks by LM, while WT cells show individual tubules connected into a loose network mostly positioned in the cell periphery (Fig. 4 B). We expected that overexpression of membrane-deforming proteins could suppress the defects of losing Dnm1 or alter mitochondrial morphology in some other way.

After visual assessment of the LM images of the 99 strains in the overexpression mini-library, we selected nine strains with mitochondrial morphologies distinct from the Δdnm1 strain, in which no additional protein was overexpressed. At LM resolution, they can be divided into three groups. Group 1 (overexpressing Ilv2, Ilv5, and Iml2) was characterized by the expansion of the clumped mitochondrial networks manifested by the appearance of a small number of cells with hyperproliferated mitochondria that occupied most of the confocal cross section area (Fig. 4 C). Group 2 (overexpressing Om14, Ptc1, and Rci50) possessed small, circular, loop-like Tom20-GFP–positive structures with diameters up to 500 nm in addition to mitochondria with Δdnm1-like morphology (Fig. 4 D). Group 3 (overexpressing Ugo1, Mmo1, and the uncharacterized protein Ydr366c) showed a partial rescue phenotype in which some cells had close-to-normal, extended mitochondria (Fig. 4 E). The strain overexpressing Ydr366c had the most pronounced rescue phenotype, with many mitochondria resembling those in WT cells.

In the second stage of the experiment, we used MultiCLEM for ultrastructural characterization of the previously selected nine strains (Fig. 4, F–H; and Fig. S5). We prepared ultrathin sections and collected high-resolution 2D EM micrographs of ~60 cell cross sections for each of the nine strains. Having EM images also allowed us to determine whether overexpression of the selected proteins caused side effects resulting in other abnormalities, which might suggest that the phenotype observed by the LM screen is unspecific. Using the data, we were able to visualize the diameters of mitochondrial tubules, their electron density, and the presence of other unusual structures in mitochondria or in the cell in general.

Most individual mitochondria in all strains had an ultrastructure similar to that in WT cells (Fig. S5 A): circular or extended structures 200–400 nm in diameter with electron density higher than that of the cytosol. In Δdnm1 cells, mitochondria are often found in large clumps.

Group 1 strains had an ultrastructural phenotype similar to the WT (Figs. 4 F and S5 B). In the Group 2 strains, we did not encounter any mitochondria that might correspond to the circular structures visible by LM: the mitochondria had neither a very large diameter comparable to the size of these structures, nor did they form any circular clumps. However, in a small number of cross sections, we observed compartments more similar to vacuoles by texture and bounding membrane ultrastructure, that were of a similar size to the structures observed by LM (Figs. 4 G and S5 C). Immunogold labeling showed that these structures do indeed contain Tom20-GFP (see next section), suggesting that the phenotype we saw by LM might have occurred as a result of increased mitophagy. Additional experiments would be required to characterize the Tom20-positive circular compartments. Due to their size, appropriate techniques might be in-resin CLEM combined with tomography or serial sectioning.

In Group 3, which partially rescued the Δdnm1 phenotype by LM, some cells overexpressing Ugo1 had mitochondria with lower electron density and also showed vacuole-like structures of irregular shape that were labeled by anti-GFP antibody (see

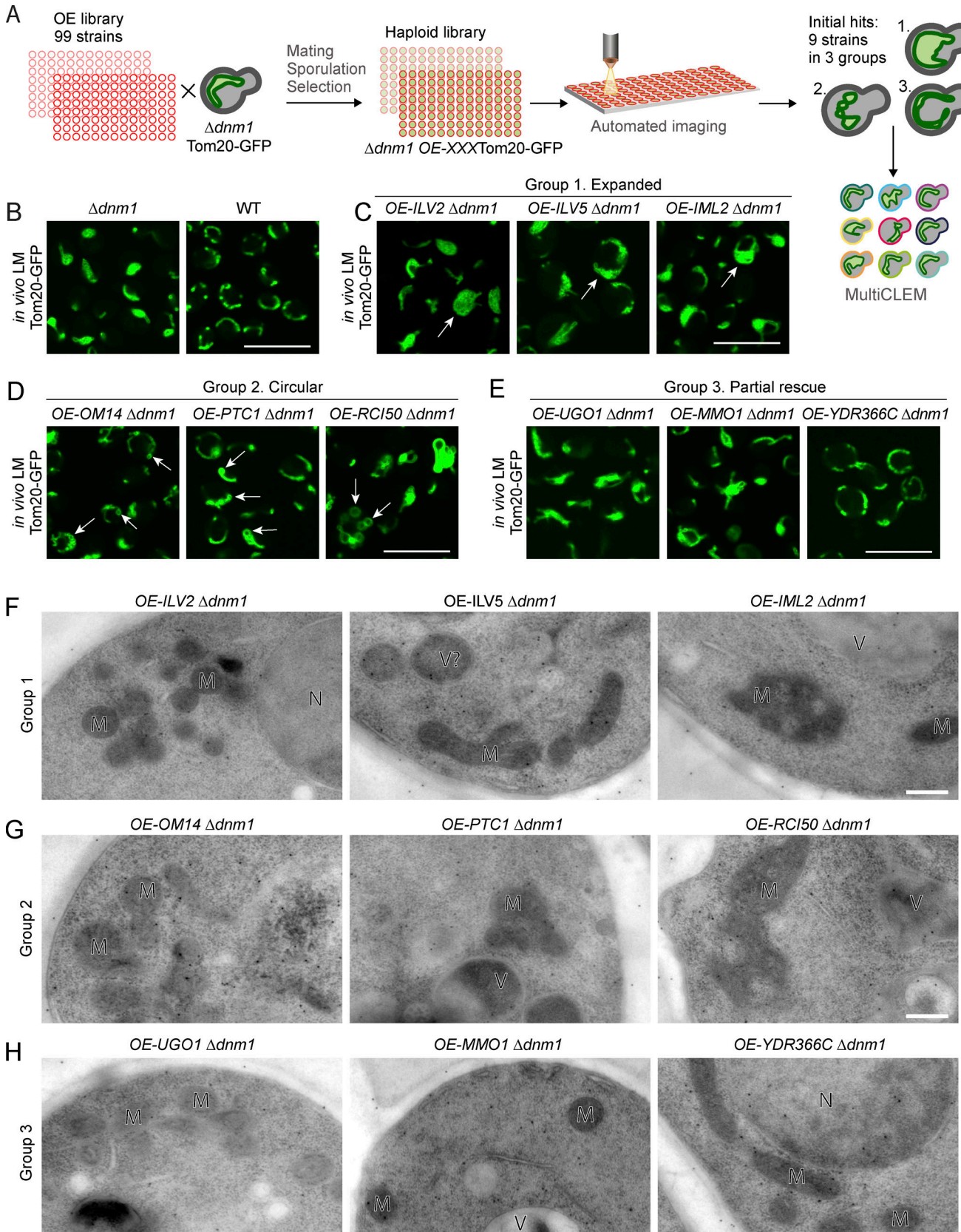

Figure 4. **Identification of new proteins affecting mitochondrial shape using two-step screening. (A)** Screen setup: using automated mating, sporu-lation, and haploid selection approaches, a tailor-made library of 99 strains was created such that all strains had a background of both Δ*dnm1* and *TOM20-GFP* (as a mitochondrial marker) as well as an overexpressed (OE) allele of one mitochondrial-associated protein (enriched for outer mitochondrial membrane

proteins). Strains were then imaged using live LM. Nine strains that had a phenotype different from Δdnm1 alone (phenotypes can be divided into three groups: 1, expanded; 2, circular; and 3, partial rescue) were selected for EM analysis using MultiCLEM. **(B–E)** Mitochondrial morphology visualized by LM in living cells expressing Tom20-GFP. **(B)** The overexpression background strain Δdnm1 TOM20-GFP compared with the WT TOM20-GFP strain. **(C)** Group 1 strains in which a subset of cells have expanded mitochondrial networks (arrows). **(D)** Group 2 strains in which circular, GFP-positive compartments are observed (arrows). **(E)** Group 3 strains in which a subset of cells have mitochondrial morphology more similar to WT, indicating a partial Δdnm1 rescue phenotype. **(F–H)** Characterization of the mitochondrial ultrastructure using MultiCLEM. F, group 1 strains; G, group 2 strains; H, group 3 strains. Organelles marked in EM images: M, mitochondria; V, vacuole; N, nucleus. (Gold beads are not specifically bound.) Scale bars: 10 µm for LM images, 200 nm for EM images.

next section). The strains overexpressing Mmo1 and Ydr366c had mitochondrial sizes, distribution, and electron density, as well as overall cell morphology, similar to that of WT (Figs. 4 H and S5 B), suggesting that overexpression of these two proteins rescues the Δdnm1 mitochondrial phenotype without side effects affecting cellular ultrastructure.

To summarize, as result of this screen in which we combined conventional LM screening and a secondary ultrastructure characterization using MultiCLEM, we identified Mmo1 and Ydr366c as potential factors that might play an additional role in establishing mitochondrial morphology. Both of them are small proteins that localize to mitochondria when tagged with GFP (Yofe et al., 2016), have an unknown function, and have predicted trans-membrane helices. We therefore changed the name of Ydr366c to Mor1 (for mitochondrial morphology).

### Multiplexed barcoding combined with GFP and immunogold labeling for organelle identification

Both mitochondria and MVBs have distinct features, making them clearly identifiable by EM; however, many organelles such as peroxisomes or small vesicles are not easily identified either by eye or computationally. Hence, to enable our method to be quantitative for a large number of traits, we sought additional means to accurately assign identity to a variety of cellular structures. Currently two ways exist to do this, immunogold labeling and correlative microscopy, and we chose to explore both approaches.

#### Immunogold labeling

We optimized an immunostaining protocol to visualize any protein fused to GFP with anti-GFP antibodies on our MultiCLEM samples. We then used this protocol to perform immunogold labeling on the set of Δdnm1 strains expressing Tom20-GFP and overexpressing the nine mitochondrial outer membrane proteins described in the previous section. This was necessary to analyze the strains overexpressing Rci50, Om14, Ptc1, and Ugo1 that had structures resembling small vacuoles in addition to mitochondria with normal morphology (Fig. S5 C). Gold beads localized to vacuolar structures in strains overexpressing Rci50, Ugo1, and Om14, supporting our suggestion that some of Tom20-GFP might be localized to vacuoles as a result of increased mitophagy.

The specificity of immunolabeling was confirmed by quantifying the number of gold beads localized to mitochondria and vacuoles in the strains where these organelles can be unambiguously identified visually. In control WT cells containing no GFP, we observed on average 0.9 gold beads per cell cross section. These were colocalized with cytosolic structures with a slight preference toward the vacuole (27% cytosol, 17% nucleus,

33% vacuole, 17% mitochondria, and 6% others; n = 500 cells). In cells with GFP-labeled vacuoles (expressing Vph1-GFP), we observed on average five beads per cell, 85% of them localized to vacuoles (n = 50 cells). In cells expressing Tom20-GFP, we observed on average six gold beads per cross section, 87% of the beads localized to mitochondria, 11% of beads were localized to the vacuole, and 3% to cytosol and other organelles (n = 60 cells). In all preparations, both WT and GFP-containing cells had a significant number of gold beads (up to 10%) localized to cell walls, which could be confidently excluded from analysis. Together, these results confirm that immunolabeling is specific and can be used with MultiCLEM to help define cellular structures.

#### Correlative microscopy

To perform in-resin CLEM, we focused on peroxisomes. Peroxisomes are small organelles with diameters smaller than the resolution of LM. We expressed a peroxisomal targeted fluorophore (Grx1-GFP-PTS1) in 14 yeast strains, each carrying a deletion of one peroxisomal protein as well as in a control strain (see Table S1 for the full list). The strains were grown on glucose-containing medium (where peroxisomes are fewer and smaller and hence harder to visualize, as a proof of principle for the effectiveness of the CLEM approach), barcoded using four Con A conjugates, and processed for EM. We successfully identified all 15 strains in fluorescent micrographs, and punctate GFP signals were visible in all strains except the Δpex22 strain, which did not correctly target this reporter to peroxisomes under these growth conditions. After correlation and analysis of 518 cells at medium and high resolution, we confidently identified 46 structures as peroxisomes in 11 strains of 14 that showed punctuate signals in LM (Fig. 5). Since we did not add fluorescent fiducial markers for correlation, we had low correlation precision and were unable to confidently correlate some signals coming from cells belonging to the three remaining strains. In the strains where we could identify peroxisomes, the peroxisome cross section diameters varied from 100 to 400 nm (Fig. 5 B), and some of the smaller peroxisomes showed increased electron density of the contents (Fig. 5 A). Peroxisomes tended to be larger in the strain lacking Ant1 (a peroxisomal ATP transporter), demonstrating the power of this approach to differentiate ultrastructural details that cannot be uncovered using conventional LM screening and image analysis.

## Discussion

In this work, we describe a new approach, termed MultiCLEM, that allows systematic, parallel, high-throughput screening for traits observable by EM. We demonstrate that parallel

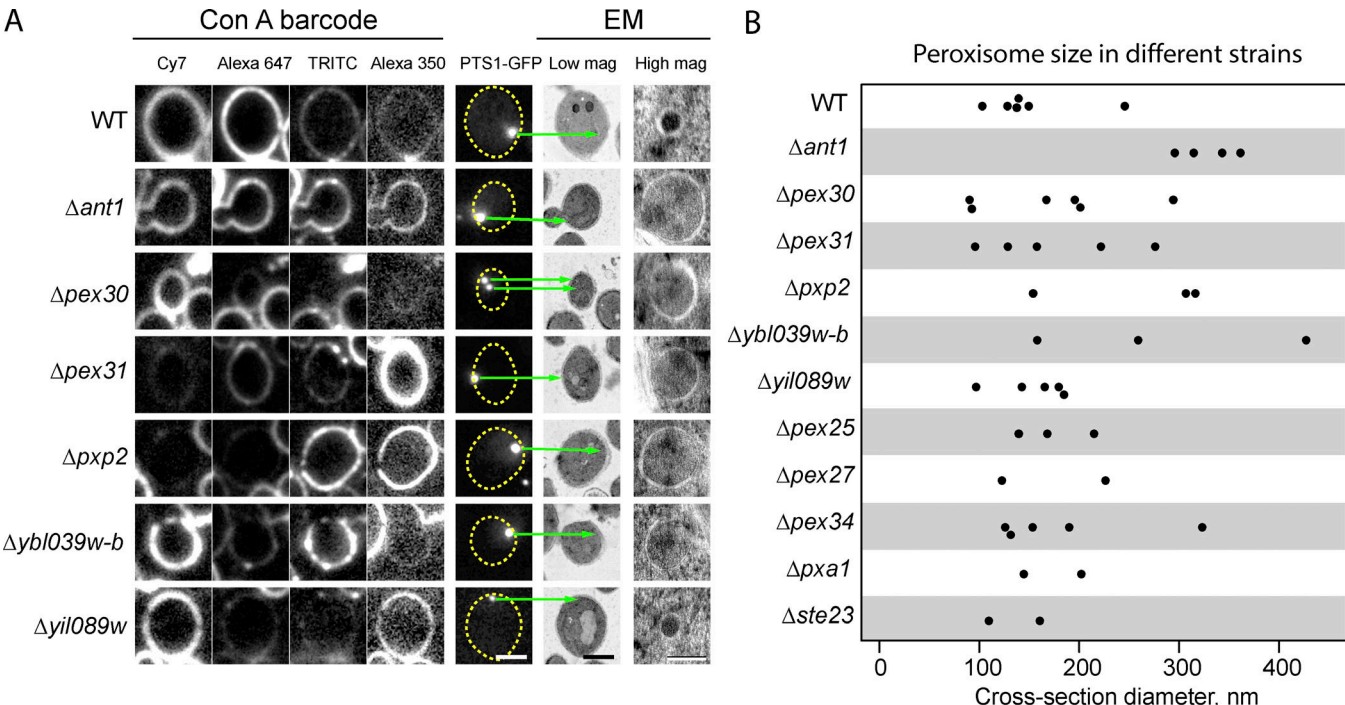

Figure 5. **Identification of peroxisomes in different deletion strains during MultiCLEM. (A)** An example set of cell cross sections in which peroxisomes could be identified in the high-resolution data. The first four columns show images taken in each fluorescent channel from which the barcode is read; the fifth column shows the GFP signal representing peroxisomes (Grx1-GFP-PTS1); the sixth column shows EM images of the corresponding cell; and the seventh column shows high-magnification (mag) EM images at the position of the GFP puncta. **(B)** Peroxisome cross section diameters measured for all identified peroxisomes in different strains. Scale bars: 2 µm for LM and low-magnification EM panels; 200 nm for high-magnification EM panel.

processing of up to 15 barcoded strains within a single EM sample can be performed on a timescale similar to that of a typical EM or in-resin CLEM experiment. A typical automated freeze-substitution processor can produce 10–20 resin blocks within 1 wk, and they can be sectioned within few days. Given two blocks per sample, 5–10 multiplexed samples can be processed and sectioned in parallel, allowing 70–140 strains to be assessed using a single freeze-substitution run. Hundreds of EM images of each strain could then be collected in an automated manner within an extended EM imaging session. Such an experiment using conventional approaches would take months of consecutive freeze substitutions and many days of laborious sectioning and EM imaging.

While demonstrating the applicability of this technique in different scenarios, we described three proof-of-principle experiments that resulted in novel insights into yeast cell biology. We described a significant variation of MVB volume fraction in strains with different ploidy and genetic background, suggesting that different strains may differentially regulate MVB biogenesis. These differences might be important for stress tolerance, though we did not observe changes in MVB volume fraction in response to changes in the osmolarity of the medium. Using MultiCLEM as part of a two-step LM–EM screening strategy, we isolated Mor1 as suppressing the Δdnm1 mitochondrial morphology phenotype when overexpressed without marked effect on mitochondrial ultrastructure. We also used GFP targeted to peroxisomes to visualize these organelles in EM in different strains grown on glucose-containing medium. Such visualization

is usually very challenging due to the small size of peroxisomes in these conditions. Here we found that peroxisomes tend to be larger in strains lacking the peroxisomal ATP transporter Ant1.

Because multiple strains are treated in one sample, all cells belonging to the different strains are prepared for EM and imaged in parallel. This eliminates any variation that arises from differences in vitrification during high-pressure freezing, sample shrinkage during freeze substitution, compression during sectioning, and contrast and brightness during post-staining and imaging. If necessary, a control sample can be included in every multiplexed experiment for direct comparison; hence our method not only accelerates data acquisition but also increases its accuracy. The parallel nature of the experiment means that quantitative comparative analysis of immunogold-labeled sections can be performed. It also facilitates automated analysis of the images, allowing direct quantitative comparison of strains using image-processing algorithms or neural networks. Multi-CLEM can be used to parallelize not only the studies of different genetic backgrounds but also conditions (such as media, stress, or drug treatments) and time points. However, the staining protocol timeline limits the time resolution of such a series to ~60 min.

In our protocol, we used five spectrally nonoverlapping fluorophores (four for barcoding and one for GFP). In principle, all five can be used for barcoding if cells contain no additional fluorescent tags. Each fluorophore had two possible staining levels (stained or not stained), giving a maximum number of

$2^5 - 1 = 31$ combinations (the combination with no staining for all fluorophores is excluded for accuracy purposes). It is reasonable to prepare five samples in parallel, which would provide for up to 155 strains. A further increase in the number of multiplexed experiments could be achieved by several means. More fluorophores with overlapping spectra could be used, and linear unmixing could be applied. Different staining levels could be used for each fluorophore, similar to existing fluorescent barcoding techniques such as BrainBow and cell fluorescent barcoding for cell sorting (Krutzik and Nolan, 2006; Weissman and Pan, 2015). Internal organelles could be stained in the same combinatorial manner, in addition to the cell wall, providing the opportunity for hundreds of samples to be barcoded at once. If that ability were to be developed, the available EM time to acquire high-resolution images of all such strains in a sample would quickly become limiting. In the future, such limitations could be overcome by increasing data collection speed using a multibeam electron microscope (Lena Eberle et al., 2015) or by precise targeting of data collection using CLEM or immunogold labeling so that only regions of interest are imaged. We note that the requirement to analyze such large amounts of EM data also presents a limitation, although we are optimistic that the rapid development of automated image analysis methods will help to overcome this. We show in this work that it is possible to use MultiCLEM to parallelize conventional CLEM experiments and visualize GFP in resin sections. The original in-resin CLEM protocol we used as a basis for MultiCLEM was used to visualize <100 GFP molecules in one diffraction-limited spot (Kukulski et al., 2011). This degree of sensitivity would be more difficult to achieve in MultiCLEM due to fluorescent background coming from the barcode staining.

We showed that our protocol adapted for budding yeast grown in suspension can work in a variety of genetic backgrounds from the common laboratory strain (BY4741) to WT isolates. However, it is possible that in some other genetic backgrounds, additional optimization of the staining procedure might be required. In principle, the MultiCLEM method can also be used for other species that can be stained with Con A, including some mammalian cell lines (Keefe et al., 2005). Similar protocols can be developed for bacterial or mammalian cells, including adherent cells, using antibodies labeled with various fluorophores, chemical dyes, or genetically encoded fluorescent proteins.

The approach that we have presented here allows both highly multiplexed experiments and parallel sample preparation. It therefore fulfills the requirements for quantitative screening of EM samples. We believe that this approach will lay the foundation for expanding systematic screening and high-throughput imaging approaches to the ultrastructural level using CLEM.

## Materials and methods

### Yeast strains
The yeast strains used in this study are listed in Table S1. The library for the peroxisomal morphology screen was prepared by mating a roGFP-PTS1 plasmid containing query strain (constructed on the basis of a synthetic genetic array–compatible strain, YMS721; Papić et al., 2013) with a collection of ~15 strains in which peroxisomal genes were deleted using a KanMx knockout cassette (Goldstein and McCusker, 1999). Automated sporulation and selection of haploids was performed using the synthetic genetic array method (Tong and Boone, 2006; Cohen and Schuldiner, 2011) in high-density format using a RoToR benchtop colony arrayer (Singer Instruments).

A similar synthetic genetic array approach was used to create the library for the mitochondrial morphology screen by mating a TOM20-GFP, Δdnm1 expressing query strain to a collection of ~100 strains in which genes for mitochondrial outer membrane proteins, and a number of other mitochondria-related proteins, have been modified to be driven by a TET-OFF promoter.

### Cell growth
For protocol development, cells were grown in YPAD medium or synthetic complete medium without tryptophan. For parallel growths of more than seven strains, we used 24-well plates. To ensure equal gas exchange rate in all wells, the plate was sealed with a gas-permeable membrane (Sigma-Aldrich) and placed on a plate shaker set to 600 rpm to achieve proper mixing of the suspension. The shaker was in turn placed in a conventional incubator at 30°C. Most experimental cultures were inoculated from colonies on agar plates, grown overnight, and in the morning diluted to $OD_{600}$ 0.1–0.2. Diluted cultures were grown to $OD_{600}$ 0.5–0.8 for assaying.

For MVB comparison, strains were grown in parallel and then either subjected to hyperosmotic shock (1 M sorbitol for 45 min) or not. For the peroxisome morphology screen, the strains were grown in synthetic defined medium supplied with monosodium glutamate and G418.

For the mitochondrial outer membrane primary fluorescent microscopy screen, the strains were transferred from agar plates into 384-well polystyrene plates for growth in liquid medium using the RoToR arrayer robot. Liquid cultures were sealed with a gas-permeable membrane and grown in a shaking incubator (Tiramax 1000, Inkubator 1000; Heidolph) overnight at 30°C in YPAD medium containing hygromycin, neurseothricin, and geneticin (G418). To drive overexpression of the TET-OFF promoter, no tetracycline was added to the medium. The strains were diluted to $OD_{600}$ of ~0.2 into plates containing YPAD medium and incubated for 4 h at 30°C. The cultures in the plates were then transferred into glass-bottom 384-well microscope plates (Matrical Bioscience) coated with Con A (Sigma-Aldrich). After 20 min, wells were washed twice with YPAD to remove nonadherent cells and obtain a cell monolayer.

For the mitochondrial outer membrane secondary EM screen, the hits from the primary fluorescent microscopy screen were grown in a 96-well plate and sealed with a gas-permeable membrane in a shaking incubator overnight at 30°C in YPAD medium. The strains were diluted to an $OD_{600}$ of ~0.2 into 96-well, 2-ml tall-well plates containing YPAD medium, sealed with a gas-permeable membrane, and incubated for 4 h at 30°C.

### Live imaging of yeast mitochondria
Strains were imaged at room temperature using VisiScope Confocal Cell Explorer system, composed of a Yokogawa spinning

disk scanning unit (CSU-W1) coupled with an inverted Olympus microscope (IX83; 60× oil objective; excitation wavelength of 488 nm for GFP and 560 nm for mCherry). Images were taken by a connected PCO-Edge sCMOS camera controlled by VisView software.

## Fluorescently labeled Con A

For cell wall staining, we purchased Con A conjugated with Alexa 350, Alexa 488, TRITC, and Alexa 647 from Life Technologies. Stock solutions with concentration 2.5 mg/ml were prepared in PBS and stored at −20°C in small aliquots. Con A conjugated to Cy7 stock solution was prepared using the following protocol (Mund et al., 2014). Sulfo-Cy7 NHS ester (Lumiprobe) was diluted in DMSO to a concentration of 10 mM. Con A (type IV; Sigma-Aldrich) 2.5 mg/ml solution was prepared in 0.2 M NaHCO$_3$ with pH 8.2. Dye and protein solutions were mixed 6:100 and incubated for 4 h at room temperature. Conjugated Con A was separated from the reaction on a disposable Sephadex G-25 column, and buffer was exchanged to PBS. The stock solution was stored frozen in small aliquots.

## Barcoding

Con A stock solutions were diluted with PBS and mixed in different combinations to yield the final staining solutions (see supplemental data for details). Yeast strains grown to logarithmic phase were pelleted by centrifugation of the multiwell plate for 5 min at 1,500 rcf. The growth medium was removed, and the pellets were resuspended in the staining solutions. The cells were incubated in the staining solutions with shaking at 30°C (the same as growth conditions) for 10 min. The cells were pelleted for 5 min at 1,500 rcf and resuspended in YPD medium. All contents of the multiwell plate were mixed together, thoroughly vortexed, and immediately processed for EM. A detailed protocol is provided in the supplemental data.

Barcoding was performed using Con A conjugated to Alexa 350, Alexa 488, TRITC, and Alexa 647. In the peroxisome CLEM experiment, we used Con A Cy7 and not Con A Alexa 488; the Alexa 488 channel was instead used to observe the GFP signal.

## EM sample preparation

We followed a standard sample preparation protocol for in-resin CLEM (Kukulski et al., 2012). Immediately after barcoding, the yeast biomass was collected using a Millipore filtering setup on a 0.45-µm nitrocellulose filter. The cell slurry was transferred to the 0.1-mm-deep cavity of a 0.1/0.2-mm membrane carrier for an HPM010 high-pressure freezing machine or Leica ICE. The cavity was covered by the flat side of a 0.3-mm carrier, and the sandwich was inserted in the high-pressure freezing machine. Resin embedding was performed using a Leica AFS2 freeze-substitution machine equipped with a processing robot. Samples were embedded in Lowicryl HM20 resin using the freeze-substitution and embedding protocol optimized for in-resin CLEM (Kukulski et al., 2012). Dry acetone with 0.1% uranyl acetate was used as the freeze-substitution medium. The blocks were trimmed with a razor blade, and 100-nm-thick sections were produced using a Diatome 35° knife on a Leica Ultracut UCT or UC7

microtome. The sections were mounted on 200 mesh copper grids with continuous carbon support film (Electron Microscopy Sciences). Grids were imaged under the fluorescence microscope the same day.

## Fluorescence microscopy for CLEM

Fluorescence microscopy was performed using a protocol for in-resin CLEM imaging (Kukulski et al., 2012). The grid was sandwiched between two coverslips with a droplet of distilled water or PBS and mounted on the microscope stage using a holder. We used a Nikon TE2000 microscope for the peroxisome morphology screen, a Zeiss Cell Observer Z1 for mitochondria morphology experiments, and a Zeiss Cell Observer HCS for all other experiments. All microscopes were controlled by their dedicated imaging software supplied by either Zeiss or Nikon. Imaging was performed at room temperature. Filter sets and other characteristics for each setup are outlined in Table S2. Usually, 5–10 positions (grid squares) were imaged on each grid. Between those, exposure times and other conditions were kept the same. After imaging, coverslips were separated, and the grid was carefully recovered and dried. If the imaging was performed in PBS, the grid was washed in distilled water before drying.

## Con A exchange analysis

To assay whether yeast cells can exchange Con A or alter their barcode in some other way when they are mixed together after barcoding, we selected two yeast strains expressing cytosolic GFP or mCherry under the *TEF2* promotor (*TEF2pr:GFP* and *TEF2pr:mCherry*). The strains were grown and stained with Con A following the procedures described above. The *TEF2pr:GFP* strain was stained with Con A conjugated to Alexa 647, and the *TEF2pr:mCherry* strain with Con A conjugated to Alexa 350 (Fig. S1 A). Cells were then prepared for EM as described above. In a typical MultiCLEM experiment, the time between pelleting of the mixed, barcoded strains and high-pressure freezing is no more than 5 min. The EM pellet was sectioned, mounted on a glass coverslip, and imaged using Zeiss Cell Observer wide-field microscope, as described above.

Additionally, we prepared samples up until the point of pelleting the mixed, barcoded strains, and then we resuspended the pellet in medium immediately (time point 0 min) or after 15, 30, or 60 min of incubation and imaged the resuspended cells at room temperature using a VisiScope spinning disk confocal microscope (described above). This live imaging experiment was performed with three biological replicates each consisting of four technical replicates. The EM sample had one biological replicate consisting of two technical replicates.

The fluorescence images were analyzed using Olympus Scan^R Analysis software in the following way. First the cells were segmented based on a merged image for all fluorescent channels. Then the cytosolic area for each cell was sub-segmented based on merged GFP and mCherry channels, and cell-wall area was sub-segmented based on merged Alexa 350 (DAPI) and Alexa 647 (Cy5) channels. The sub-segmentations were used to measure cytosolic GFP and mCherry signals and cell wall Alexa 350 and Alexa 647 signals for each cell. Then the threshold for each channel was manually selected based on the

histogram of mean fluorescent values for all cells. Based on the threshold, each cell was marked positive or negative for every channel. The majority of cells were positive for only two channels. Cells positive for both GFP and Alexa 647 or both mCherry and Alexa 350 represented the cells with correct staining pattern and were an absolute majority (Fig. S1, B and C). The cells that were assigned a "switched" color pattern, i.e., positive for GFP and Alexa 350 or mCherry and Alexa 647, upon closer examination were wrongly segmented, and no cells with such a pattern actually existed (Fig. S1, B and C). Finally, a very small proportion of cells (0.2–0.3%) were found that were positive for GFP or mCherry combined with both Con A stains (no cells were positive for both GFP and mCherry). These cells, indeed, acquired a second Con A stain after mixing and had incorrect barcodes (Fig. S1, B and C). The proportions of cells with correct and incorrect staining were very similar in all replicates and between live imaging and sectioned sample imaging experiments (Fig. S1 C). The live imaging shows that there is no cumulative Con A transfer occurring with time (Fig. S1 C).

### Immunogold labeling

Following the in-resin CLEM imaging, the grid was washed twice for 3 min in washing solution containing filtered PBS and 0.2% glycine (Sigma-Aldrich), blocked for 30 min in filtered blocking solution containing 0.5% gelatin (EMS), 0.5% BSA (MP Biomedical), and 0.2% glycine (Sigma-Aldrich) in PBS. Then samples were incubated for 2 h with AB290 rabbit anti-GFP antibody (Abcam) diluted 1:50 in filtered blocking solution, washed five times for 2 min in washing solution, blocked for 5 min in filtered blocking solution, incubated for 45 min with goat anti-rabbit 15-nm gold antibody (EMS) diluted 1:20 in filtered blocking solution, washed five times for 2 min in washing solution, and washed three times for 2 min in filtered double-distilled water. Then, grid was kept on a drop of double-distilled water until post-staining.

### EM

Prior to EM, grids were post-stained for 2 min in Reynolds lead citrate. For the mitochondria morphology screen, EM imaging was performed on a T12 Spirit Bio-Twin electron microscope (FEI) operating at 100 kV, equipped with an Eagle 2k × 2k detector (FEI). For all other experiments, EM imaging was performed on a TF30 electron microscope (FEI) operating at 120 kV, equipped with a Gatan OneView detector. SerialEM software was used for data collection (Mastronarde, 2005; Schorb et al., 2019), and the detector was operated in the full frame mode (4,096 × 4,096 pixels in the TF30 and 2,048 × 2,048 pixels in the T12). Regions imaged using LM were localized, and maps were produced at 2,000–4,000× (medium-magnification montages, corresponding to pixel size 2.5–4 nm). Complete imaging of each region (containing one grid square) required a 5 × 5 to 13 × 13 montage. The resulting montages were saved as maps in the SerialEM Navigator file and used for identification of cells during image analysis and correlation (see below). After processing, cell positions for high-resolution imaging were imported to the Navigator file and used to acquire a 2 × 2 montage of each cell with magnification 9,400× (high-magnification

images, corresponding to a pixel size of ~1 nm). Additional practical instructions for setting up the EM imaging are provided in the software manual.

### Image analysis and correlation pipeline

Here we describe the general image and data processing workflow we used. For a complete description of the software, refer to the manual provided with it. The code and documentation are available at https://github.com/ybyk/muclem and https://www2.mrc-lmb.cam.ac.uk/groups/briggs/resources (Briggs Group, 2019).

### Cell detection

We used medium-magnification montages to obtain positions and outlines of cells. Montages were blended using Blendmont software from the IMOD package (Kremer et al., 1996) to produce a single grid square overview. This overview was segmented using the pixel classification workflow in Ilastik software (Sommer et al., 2011). Ilastik was used to reliably distinguish well-preserved cells from resin, holes in resin where cells had detached, and dark regions containing cell debris. The segmentation was loaded into Matlab (Mathworks), and all processing was performed using Image Processing Toolbox functions. Cell outlines were produced by smoothing the raw Ilastik segmentation and applying watershed segmentation. Cells were automatically identified based on the object size and circularity. Outline and center coordinates for each cell were saved.

### Fluorescence intensity measurement

The Matlab control point selection tool was used to correlate LM and EM images and calculate the transformation between them. Affine transformation using 5–10 registration points was sufficient to overlay EM-derived outlines with fluorescent data and collect all signal for most cells. Cell outlines determined from the EM montages were converted to masks by a morphological dilation procedure. The intersections of masks with masks of neighboring cells and holes were subtracted from each mask to avoid measurement bias. The resulting masks were then applied to the LM images, and the fluorescent signal was measured within the masks (Fig. S2, A–D).

### Barcode determination

For each cell, median intensity was measured in each fluorescent channel within the mask. These values were normalized in two steps. First, to bring all channels to similar ranges and intensity distributions, values for each cell in each channel were normalized by subtracting the minimum observed value in this channel in all cells and dividing all values by interquartile range:

$$N_i^{green} = \frac{I_i^{green} - \min(I^{green})}{Q_3(I^{green}) - Q_1(I^{green})},$$

where $N_i^{green}$ is the normalized intensity of the $i$th cell in the green channel (as an example), $I_i^{green}$ is the raw intensity of this cell measured using the mask, $\min(I_i^{green})$ is the minimal raw intensity observed in the green channel for all cells, and $Q_1(I_i^{green})$ and $Q_3(I_i^{green})$ are the lower and upper quartiles of intensities

observed in the green channel for all cells. Interquartile range was used instead of minimum–maximum range because it better characterized the distribution shape and did not depend on outliers.

Since all cells displayed a variable total amount of labeling (but highly correlated intensity between channels), we normalized intensities the second time for each cell between channels by dividing all values for each cell by the value of the channel with maximal intensity:

$$F_i^{green} = \frac{N_i^{green}}{\max\left(N_i^{red}; N_i^{green}; N_i^{blue}; \dots\right)},$$

where $F_i^{green}$ is final normalized intensity of the $i$th cell in green channel and $(N_i^{red}; N_i^{green}; N_i^{blue}; \dots)$ are normalized intensities in red, green, blue, and other channels calculated as described above. After the second normalization, k-means clustering was performed on the final normalized intensities with the number of clusters corresponding to the expected number of color combinations.

The coordinates of cells selected for high-resolution acquisition were imported to the SerialEM Navigator file, and high-resolution EM images of these cells were collected (Mastronarde, 2005; Schorb et al., 2019). High-resolution images were blended into montages using Blendmont, and the cells were sorted according to their staining patterns. All data analysis was performed in Matlab.

### Validation of barcodes and EM preservation

For the final analysis of high-resolution micrographs in mitochondrial and peroxisomal morphology experiments, the cell barcode for each cell was verified manually after visualization in Fiji (Schindelin et al., 2012). For the MVB morphology experiment, the barcodes were not additionally verified, but the cells with poor ultrastructure preservation were excluded from analysis at the stage of MVB measurements. After completing the studies presented in this article, we developed a tool for simultaneous validation of barcodes and assessment of EM preservation. This tool facilitates the process of barcode and EM preservation quality validation. It displays fluorescent and medium resolution EM data along with all relevant information for a set of cells in a Matlab figure. The user can compare the staining patterns in each detected barcode group and confirm the quality of EM data at medium resolution with a single keyboard stroke. Cells with wrongly assigned barcode can be reassigned with a correct barcode manually. If the barcode is undiscernible or if the EM preservation is poor, the cell can be excluded from further analysis.

### MVB morphometry

Each cell micrograph was examined in IMOD without knowledge of its barcode. If an MVB was present on the cross section, it was approximated by eye as an ellipse, and its major axis and minor axis perpendicular to it (as the widest place in the direction perpendicular to the major axis) were identified as $D_1$ and $D_2$, respectively (Fig. S4 C). Area ($A$) and circumference ($C$) of each MVB cross section were calculated according to the formulas

$$A = \pi \frac{D_2 D_1}{4},$$

and

$$C = \pi \left[ 3\left(\frac{D_1}{2} + \frac{D_2}{2}\right) - \sqrt{\left(3\frac{D_1}{2} + \frac{D_2}{2}\right)\left(3\frac{D_2}{2} + \frac{D_1}{2}\right)}\, \right].$$

Areas of cell cross sections were determined from medium-magnification montages segmented using Ilastik (see above). The volume ratio of MVBs in total cell volume ($V_r$) was determined using a well-known stereology relationship between the cross section area occupied by the studied compartment and the total cell cross section area, using the areas measured above instead of using traditional stereology point counts (Howard and Reed, 2005):

$$V_r = \frac{\sum A_{MVBs}}{\sum A_{cells}}.$$

The standard error of this measurement was estimated according to the method described in Howard and Reed (2005):

$$SE = V_r \sqrt{\frac{k}{k-1}\left\{\frac{\sum u^2}{\sum u \sum u} + \frac{\sum v^2}{\sum v \sum v} + \frac{\sum uv}{\sum u \sum v}\right\}},$$

where $k$ is the total number of cross sections and $u$ and $v$ are vectors with values of cell cross section areas and MVB cross section areas for each cell, respectively. Each summation is >1 to $k$. MVB surface-to-volume area was estimated as the relation of the sum of MVB cross section areas to the sum of circumferences. Data processing and plotting were performed in Matlab, Libre Office Calc, and R (R Core Team, 2013).

### Online supplemental material

MultiCLEM barcode verification is shown in Fig. S1. Automated image processing and barcode assignment is shown in Fig. S2. A gallery of yeast cell sections from different strains and experimental conditions is presented in Fig. S3. Fig. S4 shows the ultrastructural variability of the yeast strains under normal and osmotic shock conditions. The mitochondrial morphology studied by MultiCLEM is shown in Fig. S5. Table S1 provides a list of all yeast strains used in this study, and Table S2 lists the fluorescent microscopes and filters used. A supplemental PDF describes the yeast fluorescent barcoding protocol.

## Acknowledgments

We are grateful to Markus Mund (European Molecular Biology Laboratory [EMBL], Heidelberg, Germany) and Albert Mas (Universitat Rovira i Virgili, Ramón y Cajal, Tarragona, Spain) for yeast strains and to Nadav Shai, Maria Bohnert, and Michal Eisenberg-Bord (Weizman Institute of Science, Rehovot, Israel) for sharing unpublished reagents. We thank Morgane Wartel, Lisa Maier, Einat Zalckvar, Nassos Typas, and Anne-Claude Gavin for advice and discussions. We are grateful to Michael Knop for discussions and comments on the manuscript. We

thank Martin Schorb, Yannick Schwab, and Rachel Mellwig for their help and advice on setting up the MultiCLEM method at the EMBL.

This study was technically supported by the EMBL Advanced Light Microscopy Facility, EMBL Electron Microscopy Core Facility, Weizmann Institute of Science Electron Microscopy Facility, and Medical Research Council Laboratory of Molecular Biology Light Microscopy Facility. This work was financially supported by grants from the Deutsche Forschungsgemeinschaft (SFB1129 Z2 to J.A.G. Briggs), EMBL (to J.A.G. Briggs), the Medical Research Council (MC_UP_1201/16 to J.A.G. Briggs), and the German Ministry of Education and Research (031A605 to K.R. Patil). The Schuldiner laboratory is supported by the European Research Council CoG 646604 Peroxisystem, the Deutsche Forschungsgemeinschaft (grant SFB1190 and a Deutsch-Israelische Projektkooperation [DIP] collaborative grant). N. Gabrielli was supported by the EMBL interdisciplinary postdoctoral program. M. Schuldiner is an incumbent of the Dr. Gilbert Omenn and Martha Darling Professorial Chair in Molecular Genetics.

The authors declare no competing financial interests.

Author contributions: Y.S. Bykov, conceptualization, formal analysis, methodology, investigation, writing – original draft; N. Cohen, conceptualization, formal analysis, methodology, investigation, writing – original draft; N. Gabrielli, investigation, resources, writing – review and editing; H. Manenschijn, investigation, methodology, writing – review and editing; S. Welsch, conceptualization, investigation, methodology, writing – review and editing; P. Chlanda, conceptualization, investigation, methodology, writing – review and editing; W. Kukulski, conceptualization, investigation, methodology, writing – review and editing; K.R. Patil, conceptualization, supervision, funding acquisition, writing – review and editing, project administration; M. Schuldiner, conceptualization, supervision, funding acquisition, writing – original draft, project administration; and J.A.G. Briggs, conceptualization, methodology, supervision, funding acquisition, writing – original draft, project administration.

Submitted: 17 December 2018

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
