## [Reviewer comments · The Journal of Cell Biology]

High-throughput ultrastructure screening using electron microscopy and fluorescent barcoding

Yury Bykov, Nir Cohen, Natalia Gabrielli, Hetty Manenschijn, Sonja Welsch, Petr Chlanda, Wanda Kukulski, Kiran Patil, Maya Schuldiner, and John Briggs

Corresponding Author(s): John Briggs, MRC Laboratory of Molecular Biology and Maya Schuldiner, Weizmann Institute of Science

Review Timeline:	Submission Date:	2018-12-17
	Editorial Decision:	2019-01-20
	Revision Received:	2019-05-28
	Editorial Decision:	2019-06-04
	Revision Received:	2019-06-10

Monitoring Editor: Kenneth Yamada

Scientific Editor: Melina Casadio

Transaction Report:

DOI: <https://doi.org/N/A>

January 20, 2019

Re: JCB manuscript #201812081

Dr. John A Briggs
European Molecular Biology Laboratory
Meyerhofstrasse 1
Heidelberg

Dear Dr. Briggs,

Thank you for submitting your manuscript entitled "Multiplexed electron microscopy by fluorescent barcoding allows screening for ultrastructural phenotype". The manuscript was assessed by expert reviewers, whose comments are appended to this letter. Thank you very much for your patience with the review process. We are sorry for the delay in communicating our decision to you. The two highly expert reviewers were enthusiastic about the potential of this Tools contribution, but they also listed a number of points that will need to be addressed as directly as possible and practical. We will look forward to receiving a revised manuscript from you if you can address the reviewers' key concerns, as outlined here.

These conscientious reviewers list a variety of issues that will need resolving or clarification. In particular, please provide some additional validation concerning barcode color and strain. Please also do your best to resolve each of the concerns about the quality of the documentation and technical concerns. However, unless you agree with the suggestion of Reviewer #2 to remove the demonstration as secondary screen, we would suggest leaving that in the final revised manuscript. Because of the extent of the specific issues enumerated, we will get referee input on the resubmitted manuscript but will make the final decision concerning acceptance for publication ourselves.

GENERAL GUIDELINES:

Text limits: Character count for an Tools is < 40,000, not including spaces. Count includes title page, abstract, introduction, results, discussion, acknowledgments, and figure legends. Count does not include materials and methods, references, tables, or supplemental legends.

Figures: Tools may have up to 10 main text figures. Figures must be prepared according to the policies outlined in our Instructions to Authors, under Data Presentation, <http://jcb.rupress.org/site/misc/ifora.xhtml>. All figures in accepted manuscripts will be screened prior to publication.

*****IMPORTANT:** It is JCB policy that if requested, original data images must be made available. Failure to provide original images upon request will result in unavoidable delays in publication. Please ensure that you have access to all original microscopy and blot data images before

submitting your revision.***

Supplemental information: There are strict limits on the allowable amount of supplemental data. Tools may have up to 5 supplemental figures. Up to 10 supplemental videos or flash animations are allowed. A summary of all supplemental material should appear at the end of the Materials and methods section.

The typical timeframe for revisions is three months; if submitted within this timeframe, novelty will not be reassessed at the final decision. Please note that papers are generally considered through only one revision cycle, so any revised manuscript will likely be either accepted or rejected.

Thank you for this interesting contribution to the Journal of Cell Biology. You can contact us at the journal office with any questions, cellbio@rockefeller.edu or call (212) 327-8588.

Sincerely,

Kenneth Yamada, MD, PhD
Editor, Journal of Cell Biology

Melina Casadio, PhD
Senior Scientific Editor, Journal of Cell Biology

Reviewer #1 (Comments to the Authors (Required)):

201812081 JCB review

Summary

The paper provides a new method for multiplexing electron microscopy. Since sample preparation and imaging for EM is time consuming, the authors mix multiple conditions (strains, etc.) into a single sample, identifying each condition with a barcode of fluorescent cell surface labels. They then make use of recently published tools for automated TEM imaging in SerialEM (Schorb et al., 2018, biorxiv, <https://doi.org/10.1101/389502>, which I think should be cited) to acquire images of the different experimental conditions in sections from a single sample block; the locations of cell cross-sections of each condition are identified by their fluorescent barcode in an LM, and these locations are loaded into SerialEM for automated TEM imaging, after registering LM and low mag TEM images to provide a reference map.

The central advance is the ability to distinguish multiple conditions in a mixed EM prep sample. This approach is an elegantly simple solution to a long-standing problem and, while not universally applicable, is clearly of immediate value and broad utility to the cell biology community. Personally, it would have saved me years of work and I am excited by the future possibilities. The proofs of principle demonstrate some novel cell biological insights, although they are somewhat undeveloped.

However, this is to be expected for the Tools format, and they clearly illustrate the range of possibilities for this new approach.

Overall, I am convinced of the value of the approach. I believe the data mostly demonstrate that it works, and the biological insights shown in the POPs are largely supported by the corresponding data. However, the assumption that barcode color corresponds to the correct strain should be validated. The manuscript would also be strengthened by several edits to the text. For example, a clearer and more explicit description of the steps where automation is/is not used should be included. The biological insights should also be introduced with more context and discussed in the Discussion section, even if briefly.

Below I summarise each of the main points, and highlight any issues

Introduction (p3)

"With the introduction of fully computer-controlled electron microscopes, it is now possible to perform automated and large-scale data collection."

- At this point, or in the following paragraph, it would be appropriate to briefly introduce the new automation tools available in SerialEM, or at least to cite Schorb et al., 2018, biorxiv, <https://doi.org/10.1101/389502>.

Approach; multiplexed workflow (Fig 1, Fig 2)

Label surface with barcode of fluorophores, enables conditions (strains) to be identified in mixed sample; doesn't affect cells.

Barcode reading, correlation, and targeting of EM acquisition, are automated.

- Preservation/stain is a lot more variable than other figures;
 - Low stain, suggests poor preservation (which would be confirmed at high mag), likely due to cells drying out. The EM images are not sufficient size/quality to rule this out. If preservation quality is poor in these cells, this point should be addressed in the text. See below, where I request a paragraph to describe the step for excluding cells with poor preservation that is shown in figure S1.
 - High stain, condensed cytoplasm, suggests sick cells or bad freeze.
 - Since this figure illustrates the labelling method, the authors should consider using an alternative example to demonstrate that labelling doesn't generally interfere with cell preservation, especially since cell preservation quality assessment is mentioned in the workflow in Fig S1, and appears generally good in all other figures. This point is also made on page 8, paragraph 2.
 - Alternatively, if the sample preparation protocol was altered to improve preservation quality for subsequent figures, this should be described.

"We ... assessed the reliability with which cells were assigned to the correct strain."

- The authors in fact appear to assess the reliability of cells being assigned the correct barcode color. This assumes that the barcode color as it appears in sections corresponds to the correct strain. This assumption should be validated. For example;
 - LM imaging could be done on each barcode labelled cell culture without mixing it with other cultures, maybe before, but certainly after preparation and sectioning for EM, to confirm that the barcode color is uniform.
 - Ideally, one would also test on a set of mixed strains with a variety of known ultrastructural phenotypes. This could be used to confirm that each cell has strain assignments that match when done by color or by ultrastructure. I'm not sure how a mismatch would occur if the barcode color is uniform and is unaffected by mixing strains, so I welcome the authors' and reviewers' comments on this point.

Proof of principle 1; multiplexed quantitative ultrastructural phenotyping (Fig. 2)

- Automated barcoding adjusted for this application;

1. Automated barcoding.
2. Manual screening of output 1. for quality control, barcode correction.
3. Tool to speed up step 2. developed.

- MVBs have different volume fraction in different strains. Clearly supported by data.

"While such a low false positive and negative rate may be acceptable for high-throughput analysis if the sample size is big enough, we chose to create an easy tool for quality controlling and perfecting assignments for applications where higher accuracy is desired. We created an automated tool for examination of the dataset that enables, in a few hours, assessment of hundreds of cells and manual re-assignment or exclusion."

- These sentences should be clarified, and it seems like a reference to Fig S1 is necessary. This section is focused on the accuracy of the fluorescence barcode assignment. However, quality controlling is mentioned here; is this just for barcode assignments? An automated tool for examination of the dataset is described - is this simply to present each overlaid image in the dataset to someone performing a manual visual inspection? If so the automated and manual parts of this process should be stated explicitly.

- Indeed, Figure S1 describes a step for excluding cells with poor preservation, so a short paragraph on how this is done should be added. I suggest including this in the Results text that relates to figures 1 and S1 (page 6), since the variable preservation quality in Fig 1 highlights the issue. At what point is preservation quality assessed? Is exclusion of cells because of poor preservation quality automated? - i.e. does a computer perform automated image analysis to determine preservation quality and exclude cells below a threshold? If so, how are mutant strains and other conditions that may affect apparent quality accommodated (an important question in a multiplexed sample)? Do the authors make use of their image examination tool to manually inspect and refine these exclusions? Or, if exclusion of cells due to poor preservation is not automated, do they use this tool alone to exclude cells manually? Is the gallery shown in Fig S4 before or after these exclusions? This step is not mentioned in the discussion of the time required (end of page 7).

- These points highlight an issue with Figure S1, which should be edited to indicate which steps are automated, and which steps can involve or require manual visual inspection.

- Was the assessment and manual re-assignment or exclusion step performed for the other POPs as well?

"By visual inspection, most organelles had a similar ultrastructural phenotype in all strains."

- This is rather a broad statement. Perhaps 'no obvious differences' or similar? Also, weren't vacuoles fragmented in response to the hyperosmotic treatment applied to a subset of cells? Indeed, application of this treatment should be accompanied by a description of its known effects on yeast organelle ultrastructure.

"Little is known about the MVB size, shape and abundance regulation in yeast"

- While the mechanisms are far from clear, there is at least some literature familiar to this reviewer; quantitative analysis of MVB characteristics (particularly abundance) in a range of conditions can be found in the following; Shideler et al., *Mol Biol Cell*. 2015 Apr 1; 26(7): 1345-1356; Nickerson et al., *Traffic* 2012;13:1411-1428; Russell et al., *J Cell Sci* 2012 125: 5208-5220. This section should be edited to better place the work in the context of the current literature.

- MVB size, shape and abundance are mentioned, and, while MVB volume fraction is a function of some of these characteristics, it would be nice to see them specifically shown. For example, PRICVV50 volume fraction is much lower as stated, and Fig 2B suggests the MVBs are smaller in this strain. Is their diameter indeed smaller? Are they less abundant? It would be nice to see a brief discussion of this.

Proof of principle 2, combining MultiCLEM with an upstream LM screen (p10, Fig 3)

LM screen provides 9 hits, classified into 3 phenotypes

MultiCLEM applied essentially as for POP1, to screen the 9 hits

Identified proteins that rescue loss of Dnm1 - suggests novel role for them in shaping mitochondria. Supported by data.

Makes better use of upstream LM screen since the downstream bottleneck is mitigated

- Fig 3C-E It would be helpful to title panels; eg Group 1 'expanded', Group 2 'circular', Group 3 'partial rescue'. The schematic in panel 3A does not clearly illustrate how the 9 strains fall into these groups (I could not guess which strain was in which group based on the morphology shown; maybe more stylised drawings could be done that were more representative of each class)
- Panels 3B-E higher mag images with fewer cells since phenotypes are not clear without zooming in to the PDF
- Panels 3F-H appear gold labelled, as mentioned in the Results text. This should be explained in the figure legend.
- It's a shame that Group 2 'circular' objects were not identified. While further analysis is outside the scope of the paper, could the authors suggest alternative approaches to address this question, such as serial imaging?

Proof of principle 3, combining MultiCLEM with organelle identification

p12, Fig S8, Fig 3

Identity determined by immunogold label - NB advantage of the MultiCLEM approach using HPF-FS lacking aldehyde fixation and utilising minimal EM stain (antigenicity preservation more likely)

- While I take the authors' point regarding preferential labelling of cross-section area, anti-GFP immunogold labelling on cells whose Tom20 is not tagged with GFP would be a preferable control to confirm labelling specificity. This should be done in a wt strain and at least one of the strains with GFP-containing vacuoles. Was this included in the optimization mentioned?
- Interpretation of these results is somewhat limited, and seems restricted to page 11. The conclusions from page 11 that relate to Fig 3 - that the LM phenotype is caused by increased mitophagy - could perhaps be moved or restated here on page 12.

p13, Fig 4

Identity determined by CLEM - illustrates compatibility of MultiCLEM with this technique.

- $\text{ant1}\Delta$ cells have larger peroxisomes. This is a qualitative result, with the remaining data being descriptive. The biological insight is therefore somewhat limited but the value of the method for providing compartment identity in multiple strains at once is the focus here, and is well illustrated. Moreover, there is a clear path for scaling up to a quantitative analysis.

Discussion

- Discussion of the novel biological insights presented in the proofs of principle is essentially absent. This should be added, even if briefly.
- The method has broad applicability, but I can see a few limitations and a brief discussion of some of these would strengthen this section. For example;
- The method is optimised for suspension cells, and is therefore less applicable to other cell types. Even for suspension cells, automated barcoding of the ConA fluorophore signals may need to be developed for each new cell type.
 - Authors state EM time would become limiting, but so would downstream image analysis, especially of organelles with complex morphological phenotypes.
 - In-resin fluorescence preservation is used;
 - Existing EM stains, including UA, tend to limit the intensity of fluorescence signals, even when used at the low levels of this method. Therefore CLEM of proteins at low copy number may be challenging.
 - The limited staining and conductivity of these blocks currently makes them challenging to image by 3D EM methods such as serial block face.
 - However, a bright surface label is used, and this is presumably on the cell wall, which is often

weakly stained. Therefore for applications that don't require CLEM of proteins, perhaps the EM stain could be increased.

- Could this approach also be used for time course experiments? It seems likely, although time resolution could be limited due to the time required for fluorophore treatment.

Additional Issues

p8

- Edit for clarity, eg; "while the surface to volume ratios [of MVBs] were relatively uniform (Fig. 2D)."

Fig S2

- C) Cells suitable for imaging appear white (or cream) rather than gray

- D) Blue outline is difficult to see - white or gray dotted line preferable (just overlay annotation if that's easier)

Fig S5

- Panels B and C would be clearer if titled more explicitly eg "[Cell] cross-section area, normalized" and "MVB measurement method"

- p21

- "The code and documentation is available at <https://github.com/ybyk/muclem> and <https://www2.mrc-lmb.cam.ac.uk/groups/briggs/resources>";

- I can find the documentation at the github location, but not at the /briggs/resources site. Do the authors plan to add the latter, or mirror information at the former?

Reviewer #2 (Comments to the Authors (Required)):

This is a very interesting paper that will take the (CL)EM community forward and indeed interesting to the "screening" community. The manuscript is definitely worth publishing in JCB and I only have minor comments for improvement.

In the abstract and introduction it needs to be highlighted better that the different fluorescent labels highlight different cell population but do not label / show or have any influence on the genotype / phenotype. I was initially confused why the same label ConA was used with different fluorophores.

Is anything known about dye exchange between the cells (speed)? I believe the time between mixing the cells and the point of freezing is fast enough but a short statement would be beneficial.

Top of page 6: Automated analysis is only done to medium resolution, What does that show? And is that enough to visualise the phenotype?

Second paragraph page 7, acquisition high res data: manual? Is it overselling the automated workflow a bit in that case? A bit confused whether the automated acquisition is using SerialEM or the new software. If SerialEM how does the software know which cells to acquire? Or is it working backwards, First high res EM and then association with the fluo barcode? I guess that needs a bit more clarification

Bottom page 7, specify time for this in hours days rather than refering to a paper with a technology with whom readers may not be completely familiar.

I felt the demonstration as secondary screen was not essential to the message of the paper and the multiplexing aspect is lost in that part, so no real connection to rest of the paper. I suggest a

rethink whether it is essential to the paper and doesn't take away from the (strong) message of the paper.

Top page 13, just a bit of clarification: For the correlative microscopy using GFP + multiplexing: It is almost impossible to discriminate between GFP from ConA488 so assume ConA488 is not used in that workflow.

Second paragraph page 15: needs reference what conA staining in mammalian cells looks like or even better a supplemental figure to demonstrate this.

Page 18 M&M: Alexa488 overlaps with GFP, make clear this is excluded in the GFP experiments.

I was somewhat disappointed by the ultrastructure of Fig 2A and have seen better from their published papers. I was wondering why this was. Ideally replace by better images or at least discuss.

Point by point response to editorial and reviewers concerns:

Editorial requests:

1. Character count for a Tool is < 40,000, not including spaces. Count includes title page, abstract, introduction, results, discussion, acknowledgments, and figure legends. Count does not include materials and methods, references, tables, or supplemental legends.

Title – Acknowledgements: 34133

Figure legends (Main text Figs. 1-5): 4102

Total: 38235 characters

2. Figures: Tools may have up to 10 main text figures. Figures must be prepared according to the policies outlined in our Instructions to Authors, under Data Presentation, <http://jcb.rupress.org/site/misc/ifora.xhtml>. All figures in accepted manuscripts will be screened prior to publication.

We have 5 main figures and 5 supplementary figures.

3. Supplemental information: There are strict limits on the allowable amount of supplemental data. Tools may have up to 5 supplemental figures. Up to 10 supplemental videos or flash animations are allowed. A summary of all supplemental material should appear at the end of the section.

This is now added at the end of the submitted document after the Materials and methods section

The point by point response is below. We have attached a “clean” version of our manuscript as well as one with “Track changes” for following the changes.

Reviewer #1

Introduction (p3) "With the introduction of fully computer-controlled electron microscopes, it is now possible to perform automated and large-scale data collection.". At this point, or in the following paragraph, it would be appropriate to briefly introduce the new automation tools available in SerialEM, or at least to cite Schorb et al., 2018, biorxiv, <https://doi.org/10.1101/389502>.

We have now cited the Schorb et al. 2019 paper (which was published while we were preparing this revision) at this point and elsewhere. The scripts used in our work were developed before the Python package described by Schorb et al. came out, and so we did not rely on these new tools. However, our scripts do use features introduced in SerialEM over the past few years which are described for the first time in Schorb et al. 2019 so it is indeed correct that we should reference it.

Approach; multiplexed workflow (Fig 1, Fig 2) Preservation/stain is a lot more variable than other figures;

- Low stain, suggests poor preservation (which would be confirmed at high mag), likely due to cells drying out. The EM images are not sufficient

size/quality to rule this out. If preservation quality is poor in these cells, this point should be addressed in the text. See below, where I request a paragraph to describe the step for excluding cells with poor preservation that is shown in figure S1.

- High stain, condensed cytoplasm, suggests sick cells or bad freeze.
- Since this figure illustrates the labelling method, the authors should consider using an alternative example to demonstrate that labelling doesn't generally interfere with cell preservation, especially since cell preservation quality assessment is mentioned in the workflow in Fig S1, and appears generally good in all other figures. This point is also made on page 8, paragraph 2.
- Alternatively, if the sample preparation protocol was altered to improve preservation quality for subsequent figures, this should be described.

We are aware that figure 1C showed a particularly large number of cells with poor preservation. Generally we have used four fluorophores for barcoding. We elected to use a three-fluorophore experiment for the illustrations in Figure 1 because this gives simpler schematics and because the resulting 7 colors can all be easily distinguished by eye in images. Unfortunately the only dataset we had for three colors was from an experiment assessing the effect of different electroporation methods on yeast which damages the cells, giving very poor preservation. We realize this was not wise. Since Figure 1 addresses barcoding, and to avoid the confusion here, we have simply removed panel 1C. EM preservation is much better assessed from other figures. As discussed below, we have also now better described how cell preservation is validated during our workflow on a cell by cell basis.

"We ... assessed the reliability with which cells were assigned to the correct strain." The authors in fact appear to assess the reliability of cells being assigned the correct barcode color. This assumes that the barcode color as it appears in sections corresponds to the correct strain. This assumption should be validated. For example;

- LM imaging could be done on each barcode labelled cell culture without mixing it with other cultures, maybe before, but certainly after preparation and sectioning for EM, to confirm that the barcode color is uniform.
- Ideally, one would also test on a set of mixed strains with a variety of known ultrastructural phenotypes. This could be used to confirm that each cell has strain assignments that match when done by color or by ultrastructure. I'm not sure how a mismatch would occur if the barcode color is uniform and is unaffected by mixing strains, so I welcome the authors' and reviewers' comments on this point.

Conducting a validation experiment based on a defined ultrastructural phenotype is difficult in our opinion since it would require a set of strains whose identity can be determined based on the ultrastructure observed by EM with close to 100% confidence and used as 'ground truth'. Instead we have validated the two required steps. Firstly, and most importantly, we have added a new experiment to show that barcodes are not corrupted by dye exchange, thereby validating that the barcode in sections corresponds to the correct strain. This is described in detail in the methods section, and as follows in the main text:

"We next verified that the barcodes were not corrupted during sample preparation through leaking of dyes to adjacent cells. To do this we stained a strain expressing cytoplasmic GFP with Con A conjugated to Alexa 647 and a strain expressing

mCherry with Con A conjugated with Alexa 350 (Fig. S1). We mixed these cultures together and incubated them in a pellet for 15, 30 or 60 minutes to visualize whether there is a time-dependent exchange of Con A between different cells. Then we imaged the sample immediately or after EM sample preparation. We found that, regardless of time or visualization method, less than 0.3% of cells had acquired the second cell wall stain (Fig S1; see Methods for detailed description of the analysis). Moreover, those cells that did acquire the second dye due to proximity to other cells had increased autofluorescence suggesting that they were dead or sick which would have excluded them from further analysis at later steps anyway. Since our sample preparation protocol typically requires less than 5 minutes of the cells residing in a pellet, we conclude that corruption of the barcode during sample preparation is very unlikely."

Secondly we have added error bars to the distribution of labelling intensities in figure S2E. This demonstrates that labelling is sufficiently uniform to distinguish between the presence or absence of a dye, and we have commented on this in the figure legend

"Mean normalized fluorescence intensities for each barcode identified by k-means classification of all the cells in the osmotic shock experiment, showing all 14 combinations of fluorophores used in the experiment assigned to corresponding experimental condition. Error bars show standard deviations confirming the uniformity of each barcode."

If the barcode color is uniform and is unaffected by mixing of the strains, we can see no other route to a mismatch.

"While such a low false positive and negative rate may be acceptable for high-throughput analysis if the sample size is big enough, we chose to create an easy tool for quality controlling and perfecting assignments for applications where higher accuracy is desired. We created an automated tool for examination of the dataset that enables, in a few hours, assessment of hundreds of cells and manual re-assignment or exclusion." These sentences should be clarified, and it seems like a reference to Fig S1 is necessary. This section is focused on the accuracy of the fluorescence barcode assignment. However, quality controlling is mentioned here; is this just for barcode assignments? An automated tool for examination of the dataset is described - is this simply to present each overlaid image in the dataset to someone performing a manual visual inspection? If so the automated and manual parts of this process should be stated explicitly. Indeed, Figure S1 describes a step for excluding cells with poor preservation, so a short paragraph on how this is done should be added. I suggest including this in the Results text that relates to figures 1 and S1 (page 6), since the variable preservation quality in Fig 1 highlights the issue. At what point is preservation quality assessed? Is exclusion of cells because of poor preservation quality automated? - i.e. does a computer perform automated image analysis to determine preservation quality and exclude cells below a threshold? If so, how are mutant strains and other conditions that may affect apparent quality accommodated (an important question in a multiplexed sample)? Do the authors make use of their image examination tool to manually inspect and refine these exclusions? Or, if exclusion of cells due to poor preservation is not automated, do they use this tool alone to exclude cells manually? Is the gallery shown in Fig S4 before or after these exclusions? This step is not mentioned in the discussion of the time required (end of page 7).

These questions relate to the tool for examination of the dataset, and how poorly preserved cells are excluded. Main text figure 2 (formerly figure S1) now makes clear which steps are automated, and that preservation is assessed alongside barcode assignment. We have now clarified and expanded the section pointed out by the reviewer. Now this section (page 7) reads:

“While such a low false positive and negative rate may be acceptable for high-throughput analysis, there are some situations where it is not acceptable. For such cases where higher accuracy is desired, we created an easy tool for quality controlling and perfecting assignments. Our tool enables examination of the data for each cell so that both the barcode assignment and EM preservation quality can be visually assessed at the same time. The user can then manually correct the barcode assignment or exclude cells from the dataset based on ambiguous barcode or poor EM image quality (see Methods for details). This tool enables assessment of hundreds of cells in one hour. After visual assessment the user can perform high-resolution EM only on the cells that passed the quality control.”

We have added a subsection to the materials and methods section “Image analysis and correlation pipeline” which describes and explains the use of this tool:

“Validation of barcodes and EM preservation. For the final analysis of high-resolution micrographs in mitochondrial and peroxisomal morphology experiments, the cell barcode for each cell was verified manually after visualization in Fiji (Schindelin et al., 2012). For the MVB morphology experiment the barcodes were not additionally verified, but the cells with poor ultrastructure preservation were excluded from analysis at the stage of MVB measurements. To facilitate the process of barcode and EM preservation quality validation, we developed a tool for simultaneous validation of barcodes and assessment of EM preservation. This tool displays fluorescent and medium resolution EM data along with all relevant information for a set of cells in a MATLAB figure. The user can compare the staining patterns in each detected barcode group and confirm the quality of EM data at medium resolution with a single keyboard stroke. Cells with wrongly assigned barcode can be reassigned with a correct barcode manually. If the barcode is undiscernible or if the EM preservation is poor the cell can be excluded from further analysis.”

Importantly, the gallery in Fig S4 (which is now figure S3) shows cells without any exclusions to allow the reader to assess preservation quality. We have clarified this in the legend:

“No quality control or removal of poorly preserved cells was performed on this dataset to allow the reader to assess preservation quality. All cells display good preservation.”

We mention the time required for quality assessment in the paragraph cited in the point above:

“This tool enables assessment of hundreds of cells in one hour.”

We also expand the comparison of time required for MultiCLEM and conventional EM experiments in the paragraph at the end of page 8 (formerly 7) where we add estimates of total time and manual work time. The time estimate for quality control step is added to the estimation of total time required for data processing. This paragraph now reads:

“The overall time needed to complete a MultiCLEM experiment is 2-3 weeks, similar to that required for a single EM or CLEM experiment (Kukulski et al., 2012). This similarity is due to the fact that the timeline for a single EM or a MultiCLEM

experiment is dominated by the freeze-substitution and resin-hardening step which takes 1-2 weeks in both cases. A small amount of work more than the usual EM process is required for fluorescence imaging (30 min per one grid), separate medium magnification EM imaging (2-3 h per grid requiring only 15 min of user input) and computer work (assigning strains, performing correlations and quality controlling) taking about 1 h per one grid square containing 200-300 cell cross-sections."

These points highlight an issue with Figure S1, which should be edited to indicate which steps are automated, and which steps can involve or require manual visual inspection.

As requested, we have added indications for automatic and manual steps in Figure S1 (now main text Fig. 2) and noted this in the figure legend.

Was the assessment and manual re-assignment or exclusion step performed for the other POPs as well?

The quality control tool development was finished after most of the proof-of-principle experiments were complete, so high-resolution data was collected without excluding any cells. We have included this information in a new subsection "Validation of barcodes and EM preservation" in the materials and methods which is cited above.

"By visual inspection, most organelles had a similar ultrastructural phenotype in all strains. This is rather a broad statement. Perhaps 'no obvious differences' or similar? Also, weren't vacuoles fragmented in response to the hyperosmotic treatment applied to a subset of cells? Indeed, application of this treatment should be accompanied by a description of its known effects on yeast organelle ultrastructure.

We edited the sentence as requested. I now reads:

"By visual inspection, for most organelles we observed no obvious morphology changes."

We comment on the fact that we also did not observe any changes in vacuole morphology (page 9):

"Upon the switch to hyperosmotic conditions yeast vacuoles become fragmented within a few minutes (LaGrassa and Ungermann, 2005). We did not observe any effect of changing the osmolarity of the media on vacuole morphology, though it is possible that the vacuoles started re-fusing back the barcoding step. While all EM embedding protocols may modulate ultrastructure to some extent, our approach optimizes the ability to compare strains because the control strain is processed within the same multiplexed sample."

"Little is known about the MVB size, shape and abundance regulation in yeast". While the mechanisms are far from clear, there is at least some literature familiar to this reviewer; quantitative analysis of MVB characteristics (particularly abundance) in a range of conditions can be found in the following; Shideler et al., Mol Biol Cell. 2015 Apr 1; 26(7): 1345-1356; Nickerson et al., Traffic 2012;13:1411-1428; Russell et al., J Cell Sci 2012 125: 5208-5220. This section should be edited to better place the work in the context of the current literature.

We thank the reviewer for pointing out these works. We edited the relevant paragraph in the beginning of page 10 and added more specific information on MVB biogenesis and citations of the suggested articles:

“Yeast was the main model organism used to uncover the mechanisms of MVB biogenesis, which involves fusion of individual endocytic vesicles and formation of intraluminal vesicles tightly controlled by Rab5 GTPases Vps21, Ypt52, and Ypt53 (Katzmann et al., 2001; Hanson and Cashikar, 2012; Nickerson et al., 2010; Arlt et al., 2015; Russell et al., 2012; Shideler et al., 2015). It was demonstrated that Ypt53 expression is induced under Ca²⁺ stress and this might lead to increased numbers of MVBs and promote stress-tolerance (Nickerson et al., 2012). However, variation of MVB size and abundance was not studied in detail in other growth conditions and yeast life stages.”

MVB size, shape and abundance are mentioned, and, while MVB volume fraction is a function of some of these characteristics, it would be nice to see them specifically shown. For example, PRICVV50 volume fraction is much lower as stated, and Fig 2B suggests the MVBs are smaller in this strain. Is their diameter indeed smaller? Are they less abundant? It would be nice to see a brief discussion of this.

To address this comment, we added a new plot with individual measurements of the MVB cross-section sizes as Fig. S4D and additional text on page 10:

“The reduced MVB volume fraction in SFB2 and PRICVV50 strains reflects both reduced abundance, and reduced size of the MVBs (Fig. S4D).”

Fig 3C-E It would be helpful to title panels; eg Group 1 'expanded', Group 2 'circular', Group 3 'partial rescue'. The schematic in panel 3A does not clearly illustrate how the 9 strains fall into these groups (I could not guess which strain was in which group based on the morphology shown; maybe more stylised drawings could be done that were more representative of each class)

Thank you for this suggestion - we have now added the panel titles and schematic depictions of morphologies for each of the 3 groups on Fig.3 (now Fig 4).

Panels 3B-E higher mag images with fewer cells since phenotypes are not clear without zooming in to the PDF

We enlarged the magnification of panels B-E in Fig 3 (now Fig. 4) so that the morphology is easier to see without zooming in.

Panels 3F-H appear gold labelled, as mentioned in the Results text. This should be explained in the figure legend.

These gold beads do not show any specific labeling because the immunogold labeling protocol was not optimized when these micrographs were taken. But since these micrographs represented very well the ultrastructure, we nevertheless decided to include them in the figure. We modified the figure legend in Fig. 4 (former Fig. 3) to explain this:

“(Gold beads are not specifically bound).”

It's a shame that Group 2 'circular' objects were not identified. While further analysis is outside the scope of the paper, could the authors suggest alternative approaches to address this question, such as serial imaging?

We share the reviewer's disappointment. We suggest this can be addressed using in-resin CLEM combined with tomography or, as the reviewer suggested, by serial sectioning. We introduce a sentence with this discussion at the beginning of page 13:

“Additional experiments would be required to characterize the Tom20-positive circular compartments. Due to their size, appropriate techniques might be in-resin CLEM combined with tomography or serial sectioning.”

Proof of principle 3, combining MultiCLEM with organelle identification p12, Fig S8, Fig 3. Identity determined by immunogold label - NB advantage of the MultiCLEM approach using HPF-FS lacking aldehyde fixation and utilising minimal EM stain (antigenicity preservation more likely. While I take the authors' point regarding preferential labelling of cross-section area, anti-GFP immunogold labelling on cells whose Tom20 is not tagged with GFP would be a preferable control to confirm labelling specificity. This should be done in a wt strain and at least one of the strains with GFP-containing vacuoles. Was this included in the optimization mentioned?

We have now included the requested control experiments: immunolabelling of cells containing no GFP, and those containing GFP on a vacuolar protein. These results are described in the main text as follows:

“The specificity of immunolabeling was confirmed by quantifying the number of gold beads localized to mitochondria and vacuoles in the strains where these organelles can be unambiguously identified visually. In control WT cells containing no GFP, we observed on average 0.9 gold beads per cell-cross section. These were co-localized with cytoplasmic structures with a slight preference towards the vacuole (27% cytosol, 17% nucleus, 33% vacuole, 17% mitochondria, and 6% others, N=500 cells). In cells with GFP-labeled vacuoles (expressing Vph1-GFP) we observed on average 5 beads per cell, 85% of them localized to vacuoles (N=50 cells). In cells expressing Tom20-GFP, we observed on average 6 gold beads per cross-section, 87% of the beads localized to mitochondria, 11% of beads were localized to the vacuole, and 3% to cytoplasm and other organelles (N=60 cells). In all preparations, both WT and GFP containing cells had significant number of gold beads (up to 10%) localized to cell walls which could be confidently excluded from analysis. Together these results confirm that immunolabelling is specific and can be used with MultiCLEM to help to define cellular structures.”

Interpretation of these results is somewhat limited, and seems restricted to page 11. The conclusions from page 11 that relate to Fig 3 - that the LM phenotype is caused by increased mitophagy - could perhaps be moved or restated here on page 12.

As suggested, we have restated the interpretation here:

“Gold beads localized to vacuolar structures in strains overexpressing Rci50, Ugo1 and Om14, supporting our suggestion that some of Tom20-GFP might be localized to vacuoles as a result of increased mitophagy.”

Discussion of the novel biological insights presented in the proofs of principle is essentially absent. This should be added, even if briefly.

We were concerned not to make the paper too long and to focus on the method, but we agree with the reviewer that this is an omission. We have added further interpretations throughout the text (as described elsewhere in this response), and have also added a paragraph to the discussion on page 15 that summarizes the biological insights that we present in the paper:

“While demonstrating the applicability of this technique in different scenarios, we described three proof-of-principle experiments which resulted in novel insights into yeast cell biology. We described a significant variation of MVB volume fraction in strains with different ploidy and genetic background suggesting that different strains may differentially regulate MVB biogenesis. These differences might be important for stress tolerance, though we did not observe changes in MVB volume fraction in response to changes in the osmolarity of the media. Using MultiCLEM as a part of two-step LM-EM screening strategy we isolated a gene suppressing the Δ dnm1 mitochondrial morphology phenotype without marked effect on mitochondrial ultrastructure. We also used GFP targeted to peroxisomes to visualize these organelles in EM in different strains grown on glucose-containing media. Such visualization is usually very challenging due to the small size of peroxisomes in these conditions. Here we found that peroxisomes tend to be larger in strains lacking the peroxisomal ATP transporter Ant1.”

The method has broad applicability, but I can see a few limitations and a brief discussion of some of these would strengthen this section. For example;

- The method is optimised for suspension cells, and is therefore less applicable to other cell types. Even for suspension cells, automated barcoding of the ConA fluorophore signals may need to be developed for each new cell type.

We have commented on this as requested:

“We showed that our protocol adapted for budding yeast grown in suspension can work in a variety of genetic backgrounds from the common laboratory strain (BY4741) to wild type isolates. However it is possible that in some other genetic backgrounds additional optimization of the staining procedure might be required. In principle, the MultiCLEM method can also be used for other species that can be stained with Con A, including some mammalian cell lines (Keefe et al., 2005). Similar protocols can be developed for bacterial or mammalian cells, including adherent cells, using antibodies labeled with various fluorophores, chemical dyes or genetically encoded fluorescent proteins.”

- Authors state EM time would become limiting, but so would downstream image analysis, especially of organelles with complex morphological phenotypes.

We have added:

“We note that the requirement to analyze such large amounts of EM data also presents a limitation, though we are optimistic that the rapid development of automated image analysis methods will help to overcome this.”

- In-resin fluorescence preservation is used;
 - Existing EM stains, including UA, tend to limit the intensity of fluorescence signals, even when used at the low levels of this method. Therefore CLEM of proteins at low copy number may be challenging.

We have added:

“We show in this work that it is possible to use MultiCLEM to parallelize conventional CLEM experiments and visualize GFP in resin sections. The original in-resin CLEM protocol we used as a basis for MultiCLEM was used to visualize as few

as 25 GFP molecules in one diffraction-limited spot (Kukulski et al., 2011). This degree of sensitivity would be more difficult to achieve in multiCLEM due to fluorescent background coming from the barcode staining."

- The limited staining and conductivity of these blocks currently makes them challenging to image by 3D EM methods such as serial block face. - However, a bright surface label is used, and this is presumably on the cell wall, which is often weakly stained. Therefore for applications that don't require CLEM of proteins, perhaps the EM stain could be increased.

We did not explore other sample preparation techniques to use with our approach so we think that this lies out of the scope of the discussion.

- Could this approach also be used for time course experiments? It seems likely, although time resolution could be limited due to the time required for fluorophore treatment.

We have added:

"MultiCLEM can be used to parallelize not only the studies of different genetic backgrounds but also conditions (like media, stress or drug treatments), and time points. However, the staining protocol timeline limits the time resolution of such a series to around 60 minutes."

p8 z. Edit for clarity, eg; "while the surface to volume ratios [of MVBs] were relatively uniform (Fig. 2D)."

Corrected

Fig S2 C) Cells suitable for imaging appear white (or cream) rather than gray

Corrected to "cream" in figure legend

D) Blue outline is difficult to see - white or gray dotted line preferable (just overlay annotation if that's easier)

We have added a white dotted line as suggested.

Fig S5 Panels B and C would be clearer if titled more explicitly eg "[Cell] cross-section area, normalized" and "MVB measurement method"

We add more detailed titles to this figure (now S4).

p21 "The code and documentation is available at <https://github.com/ybyk/muclem> and <https://www2.mrc-lmb.cam.ac.uk/groups/briggs/resources>";

- I can find the documentation at the github location, but not at the /briggs/resources site. Do the authors plan to add the latter, or mirror information at the former?

Both sites now contain the documentation.

Reviewer #2

In the Abstract and Introduction it needs to be highlighted better that the different fluorescent labels highlight different cell population but do not label / show or have any influence on the genotype / phenotype. I was initially confused why the same label ConA was used with different fluorophores.

We introduced edits to highlight this. The Abstract now states:

“Our approach uses combinations of fluorophores as barcodes to uniquely mark each cell type in mixed populations, and correlative light and electron microscopy (CLEM) to read the fluorescent barcode of each cell before it is imaged by EM.”

In the Introduction we added:

“The barcode provides a specific staining pattern which marks the identify of each cell but does not influence its genotype or physiology.”

Is anything known about dye exchange between the cells (speed)? I believe the time between mixing the cells and the point of freezing is fast enough but a short statement would be beneficial.

We have now performed an experiment to validate that there is no significant dye exchange between cells. This experiment is described in more detail in our reply to reviewer #1. We found that,

“... regardless of time or visualization method, less than 0.3% of cells had acquired the second cell wall stain (Fig S1A; see Methods for detailed description of the analysis)”.

Top of page 6: Automated analysis is only done to medium resolution, What does that show? And is that enough to visualise the phenotype?

Automated analysis of the medium resolution EM images is used to derive the barcode and cell coordinates, which together then permit automated high-resolution imaging. Some phenotypes can be visualized at medium resolution (for example cell cross-section area for MVB volume fraction measurements described in Methods), but other phenotypes are derived from the high-resolution images. We have clarified this section as follows:

We therefore developed a workflow for automation of barcode reading, correlation, and targeting of automated high-resolution acquisition using MATLAB and SerialEM (Mastronarde, 2005; Schorb et al., 2019)(Fig. 2). The workflow is organized in a MATLAB graphic user interface (GUI). First, the user assists correlation of medium magnification EM data with LM data, then barcodes and coordinates of all cells are automatically determined, and finally coordinates of cells are imported back to the electron microscope. SerialEM then uses these coordinates for automated high-resolution EM imaging”

Second paragraph page 7, acquisition high res data: manual? Is it overselling the automated workflow a bit in that case? A bit confused whether the automated acquisition is using SerialEM or the new software. If SerialEM how does the software know which cells to acquire? Or is it working backwards, First high res EM and then association with the fluo barcode? I guess that needs a bit more clarification.

High-resolution imaging is automated. Please see our response to the previous question where we have now clarified these points.

Bottom page7, specify time for this in hours days rather than refering to a paper with a technology with whom readers may not be completely familiar.

We have edited the paragraph in question to provide more details of the timing. It now reads:

“The overall time needed to complete a MultiCLEM experiment is 2-3 weeks, similar to that required for a single EM or CLEM experiment (Kukulski et al., 2012). This similarity is due to the fact that the timeline for a single EM or a MultiCLEM experiment is dominated by the freeze-substitution and resin-hardening step which

takes 1-2 weeks in both cases. A small amount of work more than the usual EM process is required for fluorescence imaging (30 min per one grid), separate medium magnification EM imaging (2-3 h per grid requiring only 15 min of user input) and computer work (assigning strains, performing correlations and quality controlling) taking about 1 h per one grid square containing 200-300 cell cross-sections. The increase in throughput of the most laborious steps was dramatic – fourteen samples were sectioned simultaneously (1 h for one block instead of 14 h for fourteen blocks). They were inserted into the microscope in one step and imaged in an automated manner in only two stages: a short session to acquire medium magnification maps (2-3 h) and a long session for high-magnification imaging (around 24 h). Since it is feasible to perform 2-5 such EM experiments in parallel during a 2-3 week period, this makes it feasible to now study tens or even hundreds of strains where before only a few strains could be analyzed in a similar time period.”

I felt the demonstration as secondary screen was not essential to the message of the paper and the multiplexing aspect is lost in that part, so no real connection to rest of the paper. I suggest a rethink whether it is essential to the paper and doesn't take away from the (strong) message of the paper.

We understand the reviewer's sentiments, but we believe that the idea of a secondary screen nicely highlights how our method can be applied in combination with existing screening techniques. Further, we think that the identification of YDR366C as affecting mitochondria shape is worth communicating and might not otherwise be published.

Top page 13, just a bit of clarification: For the correlative microscopy using GFP + multiplexing: It is almost impossible to discriminate between GFP from ConA488 so assume ConA488 is not used in that workflow.

The reviewer is of course correct. This is now made explicit in the methods section:

“Barcoding was performed using Con A conjugated to: Alexa Fluor 350, Alexa Fluor 488, TRITC, and Alexa Fluor 647. In the peroxisome CLEM experiment we used ConA Cy7 and not ConA Alexa 488: the Alexa 488 channel was instead used to observe the GFP signal.”

Second paragraph page 15: needs reference what conA staining in mammalian cells looks like or even better a supplemental figure to demonstrate this.

We now cite an article where Con A staining of mammalian cells can be found (Keefe et al 2005).

Page 18 M&M: Alexa488 overlaps with GFP, make clear this is excluded in the GFP experiments.

We have clarified as follows:

“Barcoding was performed using Con A conjugated to: Alexa Fluor 350, Alexa Fluor 488, TRITC, and Alexa Fluor 647. In the peroxisome CLEM experiment we used Con A Cy7 and not Con A Alexa 488: the Alexa 488 channel was instead used to observe the GFP signal.”

I was somewhat disappointed by the ultrastructure of Fig 2A and have seen better from their published papers. I was wondering why this was. Ideally replace by better images or at least discuss.

We are not sure that we understand the reviewer's question because Fig 2A (now 3A) shows a medium magnification montage where fine ultrastructure cannot be assessed. Judging by the electron density the sample preservation is acceptable. We suspect that the reviewer may have meant to refer to the poor preservation of some cells in the former figure Fig. 1C, since this point was also raised by Reviewer 1. We have addressed this point in the reply to Reviewer 1.

June 4, 2019

RE: JCB Manuscript #201812081R

Dr. John A Briggs
MRC Laboratory of Molecular Biology
Francis Crick Avenue
Cambridge CB2 0QH
United Kingdom

Dear Dr. Briggs,

Thank you for submitting your revised manuscript entitled "Multiplexed electron microscopy by fluorescent barcoding for screening of ultrastructural phenotype". You will see that the returning reviewer and we find your responses to the reviews appropriate. We are sharing some comments from the reviewer below for you to consider with respect to any final revisions, which we view as optional. We would be happy to publish your paper in JCB pending final revisions necessary to meet our formatting guidelines (see details below) and pending edits to tackle the remaining reviewer points below, as needed:

"Regarding barcode and preservation quality validation; based on the authors' responses, I believe the following to be the case:

- In the workflow (Fig. 2), there is an orange step; 'Validate barcodes and EM preservation quality'.
- The methods section describes how this step was performed (using Fiji for two POPs and at the measurement stage for POP 3) for the manuscript experiments. This section then describes the MATLAB figure tool that was developed to improve this step, but this tool was not used for the manuscript experiments.

If that's correct, I'm not concerned that the MATLAB figure tool was not used and welcome its creation, so I agree that it should be reported here. But I suggest that it would be clearer to be more explicit that it was not used for the manuscript data. I leave it to the authors how and whether they address this.

"Yeast was the main model organism used to uncover the mechanisms of MVB biogenesis ..."

- Perhaps change to "one of the main model organisms" since work from the Hopkins, Stenmark, Luzio, and others' labs would suggest this would be more appropriate."

1) Text limits: Character count for Articles and Tools is < 40,000, not including spaces. Count includes title page, abstract, introduction, results, discussion, acknowledgments, and figure legends. Count does not include materials and methods, references, tables, or supplemental legends.

2) Titles, eTOC: Please consider the following revision suggestions aimed at increasing the accessibility of the work for a broad audience and non-experts.

Title: High-throughput electron microscopy screening of ultrastructure using fluorescent barcoding

Running title (50 characters max, including spaces): Since the method is not limited to yeast we suggest making this clearer in the running title: High-throughput electron microscopy

eTOC summary: A 40-word summary that describes the context and significance of the findings for a general readership should be included on the title page. The statement should be written in the present tense and refer to the work in the third person.

Suggested eTOC: please feel free to edit while keeping with JCB style:

Bykov, Cohen, et al. describe a new method called "MultiCLEM" to screen ultrastructural phenotypes by high-throughput electron microscopy (EM) using combinations of fluorophores as barcodes to identify cells, correlative light and EM (CLEM), and EM.

3) Figure formatting: Scale bars must be present on all microscopy images, including inset magnifications. Please add scale bars to 1C, S1B, S4, S5A

4) Statistical analysis: Error bars on graphic representations of numerical data must be clearly described in the figure legend. The number of independent data points (n) represented in a graph must be indicated in the legend. Statistical methods should be explained in full in the materials and methods. For figures presenting pooled data the statistical measure should be defined in the figure legends.

Please indicate n/sample size/how many experiments the data are representative of: 3CD

5) Materials and methods: Should be comprehensive and not simply reference a previous publication for details on how an experiment was performed. Please provide full descriptions in the text for readers who may not have access to referenced manuscripts.

- Microscope image acquisition: The following information must be provided about the acquisition and processing of images:

a. Make and model of microscope

b. Type, magnification, and numerical aperture of the objective lenses

c. Temperature

d. imaging medium

e. Fluorochromes

f. Camera make and model

g. Acquisition software

h. Any software used for image processing subsequent to data acquisition. Please include details and types of operations involved (e.g., type of deconvolution, 3D reconstitutions, surface or volume rendering, gamma adjustments, etc.).

6) A summary paragraph of all supplemental material should appear at the end of the Materials and methods section.

- Please include ~1 sentence/item.

7) Conflict of interest statement: JCB requires inclusion of a statement in the acknowledgements regarding competing financial interests. If no competing financial interests exist, please include the following statement: "The authors declare no competing financial interests." If competing interests are declared, please follow your statement of these competing interests with the following statement: "The authors declare no further competing financial interests."

8) Author contributions: A separate author contribution section is required following the Acknowledgments in all research manuscripts. All authors should be mentioned and designated by their full names. We encourage use of the CRediT nomenclature.

A. MANUSCRIPT ORGANIZATION AND FORMATTING:

Full guidelines are available on our Instructions for Authors page, <http://jcb.rupress.org/submission-guidelines#revised>. **Submission of a paper that does not conform to JCB guidelines will delay the acceptance of your manuscript.**

B. FINAL FILES:

-- High-resolution figure and video files: See our detailed guidelines for preparing your production-ready images, <http://jcb.rupress.org/fig-vid-guidelines>.

Thank you for this interesting contribution, we look forward to publishing your paper in Journal of Cell Biology.

Sincerely,

Kenneth Yamada, MD, PhD
Editor, Journal of Cell Biology

Melina Casadio, PhD
Senior Scientific Editor, Journal of Cell Biology

Reviewer #1 (Comments to the Authors (Required)):

Thanks to the authors for their thorough responses. I'm satisfied with the new experiments and the other changes made; great job. In particular, the new dye exchange experiment and demonstration of barcode uniformity address key points raised by myself and reviewer #2.